# ESDMotion: SD Map Oriented Motion Prediction Enhanced by End-to-end Learning

## Abstract

Motion prediction is a crucial task in autonomous driving. Existing motion prediction models rely on high-definition (HD) maps to provide environmental context for agents. However, offline HD maps require extensive manual annotation, making them costly and unscalable. Online mapping-based methods still require HD map annotation to train the online mapping module, which is costly as well and may also suffer from the issue of out-of-distribution map elements. In this work, we explore conducting motion prediction with standard-definition (SD) maps as substitution, which are more readily available and offer broader coverage. One crucial challenge is that SD maps have low resolution and poor alignment accuracy. Directly replacing HD maps with SD maps leads to a significant drop in performance. We introduce end-to-end learning and specially tailored modules for SD maps to solve the problems. Specifically, we propose **ESDMotion**, the first end-to-end motion prediction framework that uses SD maps without any HD map supervision. We integrate BEV features obtained from raw sensor data into existing motion prediction models, with tailored designs for anchor-based and anchor-free models respectively. We find that the coarse and misaligned SD maps bring challenges to feature fusion of anchor-free model and on anchor generation of anchor-based model. Thus, we design two novel modules named Enhanced Road Observation and Pseudo Lane Expansion to address these issues. Benefiting from the end-to-end structure and new modules, ESDMotion outperforms the state-of-the-art online mapping-based motion prediction methods by 13.4% in motion prediction performance and narrows the performance gap between HD and SD maps by 73%. We will open source our code and checkpoints.

## 1 Introduction

The autonomous driving system is composed of multiple modules, including perception, motion prediction, and planning. The motion prediction module forecasts the future states of agents based on their historical data and environmental context, providing critical insights for planning and control. In providing environmental context for motion prediction models, High-Definition (HD) maps play a crucial role. These maps offer detailed and precise road geometry information, such as lane dividers, centerlines, pedestrian crossings, and stop lines, and are widely employed in motion prediction models (Gu et al., 2021; Shi et al., 2022; Zhou et al., 2022).

However, obtaining HD maps is costly. As in Fig. 1 (a), Creating HD maps requires extensive data collection and manual annotation, with updates required every 2-3 months (Li et al., 2022a). It constrains the applicability of methods that rely on HD maps. To reduce dependence on HD maps, as in Fig. 1 (b), online mapping models (Liao et al., 2023a; Li et al., 2024b; Yuan et al., 2024) are developed. These models predict HD map elements around the vehicle using sensor data in real time, providing input for downstream modules. Nevertheless, the training of these supervised models still depends on ground-truth annotations, facing the challenge of annotation and generalization.

By comparison, standard-definition (SD) maps, like those provided by Google Maps and Baidu Maps, are available at low cost and cover a wide range of areas (Jiang et al., 2024). Commonly used in human driving, SD maps provide information on road direction and intersection structures, and assist in route planning and driving maneuvers like turning. However, compared with HD maps, SD maps have two major weaknesses: (1) **Low Resolution**. SD maps only indicate the general

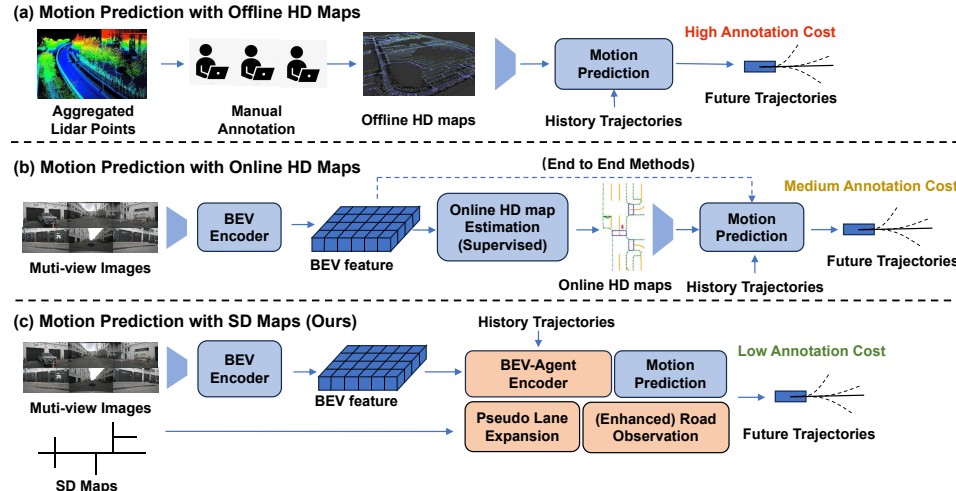

Figure 1: **Comparison of Different Map Usage Approaches for Motion Prediction.** (a) Offline mapping methods require extensive annotations, making them costly and unscalable. (b) Online mapping methods use the output of map estimation models as input for motion prediction or employ an end-to-end approach. These methods still need some HD map annotations for supervised training. (c) We use SD maps as the only map information in an end-to-end framework and achieve performance close to HD maps.

direction of roads without providing lane-level details. A road may be represented by just one or two polylines in SD maps. (2) **Poor Alignment**. Due to localization errors, the polylines in SD maps may not align with the center of the roads, and could even be outside the roads. **Our initial experiments show that directly substituting SD maps for HD maps in motion prediction tasks results in a significant drop in performance**.

The development of end-to-end autonomous driving architectures (Hu et al., 2023; Jiang et al., 2023) presents an attractive way to capture task-specific information directly from raw sensor data. These features encompass environmental information around the agent, serving a similar function to maps, thereby reducing reliance on map precision. During the process, SD maps could serve as a rough guide for the feature aggregation, providing an understanding of the road's general layout, which could potentially achieve performance comparable to using HD maps with detailed road information.

Based on the motivation, we propose **ESDMotion**, an end-to-end motion prediction framework that uses SD maps without any HD map input or supervision, as in Fig 1 (c). The framework is compatible with both anchor-based and anchor-free motion prediction models (We use DenseTNT (Gu et al., 2021) and HiVT (Zhou et al., 2022) in this paper). To construct the end-to-end model for SD maps, we extract BEV features from raw data with an encoder, and incorporate several modules, including a BEV-agent encoder, BEV-lane encoder, and Road Observation into base motion prediction models, to efficiently fuse BEV features with agent features and SD map features.

We point out two problems caused by the usage of SD maps on anchor-based and anchor-free models and design new modules to address them. Anchor-based models like DenseTNT select candidate goal points (anchors) form maps. The low resolution and poor alignment accuracy results in poor distributions of goal points, for example, there may be no goal point around agents. We solve this issue by introducing an anchor generation method called **Pseudo Lane Expansion**, which generates extra pseudo anchors parallelly based on original SD map instances to improve anchor distributions. For anchor-free models such as HiVT, due to sparse and misaligned SD maps, limited or unhelpful BEV features could be fused with SD map features in our original Road Observation module which uses standard deformable attention. Thus, we modify the multi-head deformable attention by adding reference points and introducing a head weighting mechanism. We propose **Enhanced Road Observation** to sample a wider range of BEV features around SD instances.

Benefit from the end-to-end architecture and modules specially designed for SD maps, our models achieves superior accuracy with SD maps as the only map information, surpassing state-of-the-art online HD mapping-based motion prediction models (Gu et al., 2024a;b) with a reduction of 13.4% (minADE) and 7.8% (minFDE) on anchor-free model. We also pay attention to the performance gap

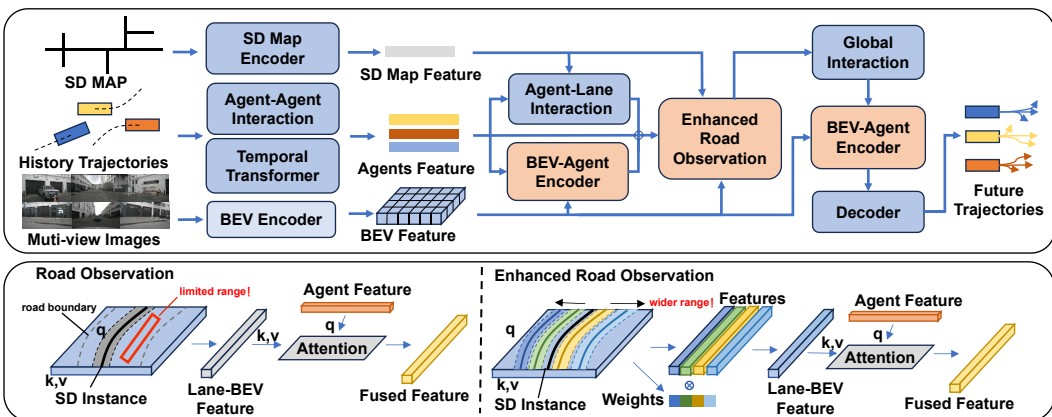

Figure 2: **HiVT-based Model**. Upper: Overall structure. The modules colored in orange are proposed by us. The inputs are first processed through the original encoding modules of the HiVT model and a BEV encoder to extract features. These features are then passed through the BEV-Agent encoder and (Enhanced) Road Observation we designed to obtain aggregated features for each agent. Global Interaction enables information exchange between agent features, and the decoder predicts future trajectories. Lower: Road Observation VS. Enhanced Road Observation. Due to the low resolution of SD maps, the coverage of Road Observation using standard deformable attention is limited. Enhanced Road Observation, by extending reference points along parallel lines, allows sampling of BEV features over a larger area.

between SD maps and HD maps. For our HiVT-based model, the performance gap between using SD and HD maps is reduced by 73% (minADE) and 44% (minFDE) compared to the original HiVT model. For DenseTNT-based model, the reductions are 77% and 84%. We further analyze how the base model and end-to-end architecture influence this gap. Our contributions are threefold:

- We propose an SD map oriented end-to-end motion prediction framework, achieving superior accuracy compared to online HD map based motion prediction models.

- We introduce a BEV-SDmap interactor called Enhanced Road Observation, and a goal point generation method Pseudo Lane Expansion to improve performance with SD map.

- We analyze the factors affecting the performance gap between SD and HD maps, including the type of base model and the application of end-to-end architecture.

Due to limited space, we discuss more details of **Related Works** in appendix A.

## 2 METHOD

In this section, we first introduce the structure of ESDMotion for anchor-free and anchor-based models in Sec. 2.1.1 and modules we merge into base models in Sec. 2.1.2. Then, we point out two problems caused by the usage of SD maps on anchor-free and anchor-based models and propose our solutions. For the anchor-free model, we enhance the feature fusion with SD map through Enhanced Road Observation in Sec. 2.2. For the anchor-based model, we improve anchor distribution from the SD map with Pseudo Lane Expansion in Sec. 2.3.

### 2.1 E2E STRUCTURE FOR MOTION PREDICTION WITH SD MAPS

#### 2.1.1 OVERALL STRUCTURES

To reduce the performance gap between SD maps and HD maps, we extend motion prediction models in an end-to-end manner. We use an existing encoder (Li et al., 2022b) to obtain BEV features from raw sensor data. Then we propose modules including BEV-Agent encoder, BEV-Lane encoder, and Road Observation for feature fusion and we incorporate them into appropriate positions in anchor-free and anchor-based models.

**The anchor-free model.**  For anchor-free model HiVT (Zhou et al., 2022), the feature of each agent is obtained from original agent-agent interaction and temporal transformer. Then, the agent features are enhanced with the SD lane feature, BEV feature, and BEV-lane feature in Agent-Lane interaction, BEV-Agent Encoder, and Road Observation module. After a global interaction that exchanges the information among agents, BEV-agent fusion is applied again. Finally, a decoder generates future trajectories from the agent features. The process is shown in Fig. 2.

**The anchor-based model.**  For anchor-based model DenseTNT (Gu et al., 2021), the sparse goal points is generated from maps first. The features of SD maps and agents are extracted by a sparse context encoder, which is based on VectorNet (Gao et al., 2020) and enhanced with BEV-Agent encoder and BEV-Lanes. The sparse goal is scored and selected with extracted features. Then, the dense goal candidates are generated from selected sparse ones in the dense goal encoder and their probability distribution is computed. Finally, a decoder predicts target points based on the distribution and decodes the future trajectory. The process is shown in Fig. 3.

### 2.1.2   SD Map Oriented E2E Modules

**BEV encoder.**  We adopt BEVFormer (Li et al., 2022b) as the BEV encoder, which extracts features from multi-view images using an image backbone (e.g., ResNet50). Then, it uses BEV2PV look-up to construct BEV features $B \in \mathbb{R}^{H \times W \times C}$ with the intrinsic and extrinsic of each camera. Note that other BEV encoders could work as well, for example, LSS (Philion & Fidler, 2020).

**BEV-Agent encoder and BEV-Lane encoder.**  Since BEV features are spatial, it is common to sample BEV features by positions of agent or lane and apply feature fusion. Previous study (Gu et al., 2024b) explores the fusion of agent features with BEV features in motion prediction models. It uses an approach similar to the Vision Transformer (ViT) (Dosovitskiy, 2020) to handle offline BEV features. In the method, BEV features are projected into coarse patches, with the patch corresponding to the agent's location serving as the query, and other patches acting as keys and values in an attention mechanism. The resulting features are then concatenated with the agent features. However, the projection can lead to the loss of fine-grained details, and the simple concatenation could limit the direct interaction between the two types of features.

To this end, we propose a BEV-Agent encoder that uses deformable attention (Zhu et al., 2020) for more direct and efficient feature fusion. We donate BEV feature $B \in \mathbb{R}^{H \times W \times C}$, agent feature $Q_A \in \mathbb{R}^{N \times D}$, and 2D positions of agents $p \in \mathbb{R}^{N \times 2}$, where N represents numbers of agents and D denotes the hidden dimension. BEV-Agent encoder fuse feature through:

$$F_{\text{BEV-Agent}} = \text{DeformAtt}(Q_A, T_{\text{ref}}(p), B) \tag{1}$$

Where $T_{\text{ref}}$ is the translation from ego vehicle coordinate system to BEV grids, $F_{\text{BEV-Agent}}$ is the resulting BEV-Agent feature. The BEV-Lane encoder is similar to the BEV-Agent encoder. It just replaces agent features with SD lane features. These two encoders sample BEV features spatially close to the agent or lane, and fuse it with the agent or lane feature, enabling the agent or lane to capture information about its surroundings from raw data.

**Road Observation.**  The Road Observation module applies integration among the BEV features, SD map features, and agent features. It first uses a BEV-Lane encoder to get BEV-SD map features, then fuses them into agent features via attention. These operations enrich SD map features with visual information and interact them with agent features, providing agents with the visual and geometrical context of roads and thus named Road Observation. Specifically, We denote the vectorized SD map as $m \in \mathbb{R}^{N_m \times 2}$, with the encoded map features represented by $F_{map} \in \mathbb{R}^{N_m \times D}$, where $N_m$ is the total number of points constituting the polylines. We obtain the BEV-SDmap feature via

$$F_{\text{BEV-SDmap}} = \text{DeformAtt}(F_{map}, T_{ref}(m), B) \tag{2}$$

and then obtain the fused features from agents through

$$F_{\text{fused}} = \text{Atten}(Q_A, F_{\text{BEV-SDmap}}, F_{\text{BEV-SDmap}}) \tag{3}$$

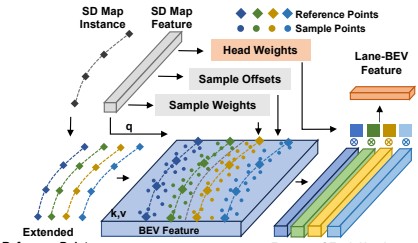

Figure 3: **DenseTNT-based Model.** Left: Overall structure. The sparse context encoder, enhanced with BEV-agent encoder and BEV-lane encoder, extracts features from both the agent and the SD map, while the dense goal encoder densely samples target points and generates a probability distribution. The decoder predicts target points based on this distribution and decodes the future trajectory. Right: Original SD map vs. Pseudo Lane Expansion. Target points generated directly from the SD map may not be near the agent, while Pseudo Lane Expansion generates pseudo target points along the road direction near the agent.

## 2.2 ENHANCE FEATURE FUSION WITH SD MAP IN ANCHOR-FREE MODEL

**Challenges in feature fusion with SD maps.** In the Road Observation, we utilize deformable attention to fuse SD map features with BEV features, generating road features. The 2D coordinates of the polylines in the vectorized map are selected as reference points. Standard deformable attention initializes sample points around these map points, sampling BEV features near the polylines. This process works well with HD maps whose polylines are accurate and dense. The sampling area of HD map lines almost covers the entire road, providing information of a wide range.

However, using SD maps as input introduces challenges. Due to low resolution, SD maps contain fewer polylines and reference points. The sample points are in a limited region and can not cover the entire road, as shown in Fig. 2 (lower left). Even worse, the misaligned SD lines may fuse useless BEV features around it. For example, it may capture features far away from the road.

**Enhanced Road Observation.** We observe that SD maps accurately indicate road direction despite alignment errors. Thus, we aim to expand reference points by creating parallel lines. In this way, a single line gets a wide sample area. Even if the original line from the SD map is out of the road, some of its parallel lines may still be within the road and sample useful BEV features. Based on this insight, we propose **Enhanced Road Observation**. We denote the SD map $S \in \mathbb{R}^{N_s \times N_d \times 2}$ with $N_s$ SD map instances and each instance consists of $N_d$ points. For $S_i$, we generate the extended reference points $\hat{S} \in \mathbb{R}^{N_s \times N_l \times N_d \times 2}$ through:

$$\hat{S}_{ij} = \text{Parallel}(S_i, l_j) \tag{4}$$

For a single SD map instance $S_i$, a set of parallel lines $\{\hat{S}_{i1}, \hat{S}_{i2}, \ldots, \hat{S}_{iN_l}\}$ are generated. Where $N_l$ is the number of parallel lines and $l_j$ represents the distance between the original polyline and each parallel line. To simplify processing, the number of parallel lines $N_l$ is set equal to the number of heads $N_h$ in the deformable attention mechanism.

Next, we modify deformable attention to accommodate multiple sets of reference points. In standard deformable attention, multiple heads share a single set of reference points, while **we assign each head with its own set of reference points corresponding to a parallel line**. Since not all heads extract meaningful features (e.g., some parallel lines may lie outside the road), we apply a learnable weight parameter $h_j$ to the features obtained by each

Figure 4: **Enhanced Road Observation.** We extended several sets of reference points for deformable attention and predict weight for each set.

head:

$$F_{\text{BEV-SDmap}} = \sum_{j=1}^{N_h} h_i W_j \left[ \sum_{k=1}^{K} A_{i,j,k} \cdot W_j' B \left( \hat{s}_{i,j} + \Delta \hat{s}_{i,j,k} \right) \right] \tag{5}$$

Where $K$ denotes the number of sample points. $W$ and $W' \in \mathbb{R}^{D \times D}$ are linear projections applied to the BEV features $B$. The sample points $A \in \mathbb{R}^{N_d \times N_h \times K}$, sampling offsets $\Delta \hat{s} \in \mathbb{R}^{N_d \times N_h \times K}$, and head weights $h$ are all obtained through different linear projections from the SD map features $Q_s$, which serve as the query. As in standard deformable attention, the weights $A$ are normalized using softmax along the last dimension. We demonstrate the process as in Fig. 4.

For the head weights $h$, multiple parallel lines from an SD instance may capture valuable environmental information, and softmax would overly prioritize a single head. DQNv4 (Xiong et al., 2024) discusses the need for softmax normalization in attention and claims that normalization becomes unnecessary when the degradation issue does not exist. Thus, we do not use functions such as Sigmoid of Softmax to normalize head weights.

## 2.3 IMPROVE ANCHOR DISTRIBUTION FROM SD MAP IN ANCHOR-BASED MODEL

**Challenges in anchor generation with SD maps.** In the anchor-based model DenseTNT, the challenges of using SD maps lie in anchor generation. The model densely samples points around candidate target points from a vectorized map, then predicts the probability for each, selecting the final target point based on these probabilities. As a result, the map has a direct impact on the distribution of candidate target points. Unfortunately, due to the low resolution and alignment accuracy of SD maps, there may be no candidate points near the agent or its future trajectory, which significantly reduces prediction accuracy. Fig. 3 shows the matter.

**Pseudo Lane Expansion.** To address this issue, adding extra pseudo anchors is a straightforward way. Since DenseTNT is designed around lane and goal features, we directly input the expanded SD lines into the model. We denote a single polyline in the SD map containing $N_d$ 2D points $p \in \mathbb{R}^{N_d \times 2}$, the unit normal vector of the vector from the i-th point to the (i+1)-th point $\mathbf{n_i}$. The i-th point $\hat{p}_{ji}$ of the j-th expanded line is calculated as:

$$\hat{p_{ji}} = p_i + d_j \mathbf{n_i} \tag{6}$$

Where $d_j$ is the distance between the j-th expanded line and the original line. Because DenseTNT's original lane scoring or goal scoring modules predict weights for each polyline or target point, we no longer predict weights for each parallel line as Enhanced road observation. This simple but effective anchor generation method greatly improves prediction performance with SD maps. We demonstrate the process as in Fig. 3 (right).

**Adaptive Pseudo Lane Expansion.** Because of the variability in road structures and the distribution of SD maps, using fixed parameters, including the number and distances of the extended parallel lines in Pseudo Lane Expansion, often fails to achieve optimal performance across diverse scenarios. Thus, we adapt the parameters based on the distance of the SD map lane relative to the vehicle to predict, and the density of SD map lanes. **(1) Distance.** If the closest distance between the SD map and the target vehicle is large, it likely indicates poor alignment of the SD map. In such case, we generate more pseudo lanes on the side closer to the vehicle while reducing the number on the opposite side to decrease the creation of irrelevant lines, as in Fig. 5 (left). **(2) Density.** For sparsely distributed SD lanes (only one or two SD lanes in the range), we increase the number of pseudo lanes to ensure better coverage of the road. Conversely, in dense areas (e.g., intersections), we decrease the number and spacing of pseudo lanes to avoid overlap and interference as in Fig. 5 (right). Furthermore, we design a set of pseudo lanes for special situation that there is no SD lanes in the range. The detail is discussed in C.2.

## 3 EXPERIMENTS

### 3.1 DATASET AND METRICS

**Dataset.** We conduct experiments on the nuScenes dataset (Caesar et al., 2019), which contains 1,000 driving scenes of approximately 20 seconds each. The dataset includes various sensor data

Table 1: **Quantitative Results of ESDMotion.** Maps indicates the source of map inputs. "BEV" and "E2E" indicate the usage of BEV feature and end-to-end structure. "ESDMotion++" means ESDMotion enhanced by Adaptive Pseudo Lane Expansion. The results of Base/Unc/BEVPred methods with GT maps is the same as "Offline Map".

| Base Prediction Model | | | | | HiVT (Zhou et al., 2022) | | | DenseTNT (Gu et al., 2021) | | |
|---|---|---|---|---|---|---|---|---|---|---|
| Method | Map Model | Map Type | BEV | E2E | minADE↓ | minFDE↓ | MR↓ | minADE↓ | minFDE↓ | MR↓ |
| Offline Map | GT | HDmap | × | × | 0.3868 | 0.8063 | 0.0870 | 0.8809 | 1.4890 | 0.1903 |
| Offline Map | GT | SDmap | × | × | 0.3998 | 0.8207 | 0.0918 | 1.2117 | 1.9849 | 0.2776 |
| Base Online Map | MapTR | HDmap | × | × | 0.4234 | 0.8900 | 0.0955 | 1.0462 | 2.0661 | 0.3494 |
| Base Online Map | MapTRv2-CL | HDmap | × | × | 0.3657 | 0.7473 | 0.0710 | 0.7664 | 1.3174 | 0.1547 |
| Base Online Map | MapTRv2 | SDmap | × | × | 0.4429 | 0.9165 | 0.0986 | 1.3692 | 2.2417 | 0.3937 |
| Unc (Gu et al., 2024a) | MapTR | HDmap | × | × | 0.4036 | 0.8372 | 0.0822 | 1.1190 | 2.1502 | 0.3669 |
| Unc (Gu et al., 2024a) | MapTRv2-CL | HDmap | × | × | 0.3588 | 0.7232 | **0.0660** | 0.8123 | 1.3426 | 0.1567 |
| Unc (Gu et al., 2024a) | MapTRv2 | SDmap | × | × | 0.4285 | 0.9007 | 0.0956 | 1.3020 | 2.1364 | 0.3738 |
| BEVPred (Gu et al., 2024b) | MapTR | HDmap | ✓ | × | 0.3617 | 0.7401 | 0.0720 | 0.7608 | 1.4700 | 0.2593 |
| BEVPred (Gu et al., 2024b) | MapTRv2-CL | HDmap | ✓ | × | 0.3652 | 0.7323 | 0.0710 | 0.7630 | 1.3609 | 0.1576 |
| BEVPred (Gu et al., 2024b) | MapTRv2 | SDmap | ✓ | × | 0.3904 | 0.7690 | 0.0741 | 1.1940 | 2.0029 | 0.3285 |
| ESDMotion(Ours) | GT | SDmap | ✓ | ✓ | **0.3134** | **0.6662** | 0.0737 | 0.7941 | 1.3863 | 0.1627 |
| ESDMotion++(Ours) | GT | SDmap | ✓ | ✓ | **0.3134** | **0.6662** | 0.0737 | **0.7597** | **1.3105** | **0.1523** |
| ESDMotion++(Ours) | MapTRv2 | SDmap | ✓ | ✓ | 0.3147 | 0.6671 | 0.0740 | 0.7712 | 1.3260 | 0.1561 |
| ESDMotion++(Ours) | MapTRv2-CL | HDmap | ✓ | ✓ | 0.3114 | 0.6692 | 0.0727 | 0.7529 | 1.3088 | 0.1517 |
| ESDMotion++(Ours) | MapTR | HDmap | ✓ | ✓ | 0.3176 | 0.6738 | 0.0751 | 0.7616 | 1.3139 | 0.1533 |

such as camera inputs, annotations, and high-definition (HD) maps. While the dataset itself does not provide standard-definition (SD) maps, we follow (Jiang et al., 2024) to extract SD maps from OpenStreetMap (Haklay & Weber, 2008) and align them with the coordinate system of nuScenes . We strictly follow the benchmark protocol in state-of-art online mapping-based motion prediction methods (Gu et al., 2024a;b). We use trajdata (Ivanovic et al., 2024) interface to access the past and future trajectories of agents. We upsample the trajectories to 10Hz and predict 3 seconds of future trajectories from 2 seconds of past trajectories. Only samples with complete past and future trajectories are used for training and evaluation.

**Metrics.** We follow the metrics in (Gu et al., 2024a;b) as well, with three widely used metrics: minimum Average Displacement Error (minADE), minimum Final Displacement Error (minFDE), and Miss Rate (MR). We predict multiple future trajectories and calculate the ADE and FDE for the trajectory closest to the ground truth, referred to as minADE and minFDE. ADE measures the average $L_2$ distance between predicted and ground-truth trajectories, while FDE measures the $L_2$ distance between the final points of the predicted and ground-truth trajectories. Miss Rate is the proportion of samples where FDE exceeds a certain threshold.

## 3.2 RESULTS

**Competitive Performance.** We compare ESDMotion mainly with online mapping-based motion prediction methods. We select MapTR(Liao et al., 2023a) and MapTRv2-CenterLine(Liao et al., 2023b) as online map estimation models. As shown in Table 1, both anchor-free and anchor-based model achieves significant improvements. Our anchor-based model reduced minADE and minFDE by 13.4% and 7.8%.

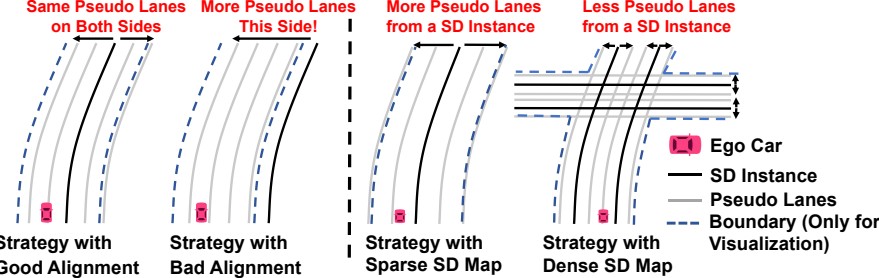

Figure 5: **The Strategy of Adaptive Pseudo Lane Expansion.**

Table 2: **Performance Gap Between HD Maps and SD Maps.** We use different maps as input for the offline mapping method and ESDMotion. In general, ESDMotion narrows the gap.

| Base Prediction Model | | | | HiVT (Zhou et al., 2022) | | | DenseTNT (Gu et al., 2021) | | |
|---|---|---|---|---|---|---|---|---|---|
| Method | E2E | HD map | SD map | minADE↓ | minFDE↓ | MR↓ | minADE↓ | minFDE↓ | MR↓ |
| Offline Map | ✗ | ✓ | ✗ | 0.3868 | 0.8063 | 0.0870 | 0.8809 | 1.4890 | 0.1903 |
| | ✗ | ✗ | ✓ | 0.3998 | 0.8207 | 0.0918 | 1.2117 | 1.9849 | 0.2776 |
| | ✗ | ✗ | ✗ | 0.4192 | 0.8727 | 0.1018 | - | - | - |
| ESDMotion++ | ✓ | ✓ | ✗ | 0.3099 | 0.6581 | 0.0721 | 0.7166 | 1.2752 | 0.1483 |
| | ✓ | ✗ | ✓ | 0.3134 | 0.6662 | 0.0737 | 0.7597 | 1.3105 | 0.1523 |
| | ✓ | ✗ | ✗ | 0.3502 | 0.7765 | 0.1051 | - | - | - |

**Performance Improvement from End-to-End Method.** As illustrated in Table 1, compared to base models (HiVT and DenseTNT) with SD maps as input, the end-to-end approach improves performance. With features extracted from raw data, it reduces minADE by 21.6% and 37.8% respectively on anchor-free and anchor-based models and decreases minFDE by 18.8% and 34.0%.

**Performance gap between HD maps and SD maps.** We examine how the type of base models and the usage of end-to-end framework influence the performance gap between HD maps and SD maps on our anchor-based and anchor-free model. We show the results in Table 2. **For base motion prediction models, the anchor-based model DenseTNT is affected more greatly by the precision of the maps than the anchor-free model HiVT.** Compared within ESDMotion, the minADE gap of maps on the anchor-based model is 0.0431(5.7%), while on the anchor-free model it is just 0.0035(1.1%). Compared with HD map based online methods in Table 1, the anchor-free model achieves larger improvement than the anchor-free model. We speculate that this is due to the different ways these models utilize and depend on maps. Anchor-free model HiVT are designed around agent features, with map features serving as auxiliary information, integrated through cross-attention. This makes the model less dependent on the map, allowing it workers even without any map input. Additionally, the robust map encoder and cross-attention mechanism can still generate useful map features from lower-precision SD maps. As a result, the performance gap between HD and SD maps is relatively small. In contrast, anchor-based model DenseTNT directly uses vectorized maps to extract goal points, predict their probability distribution, and select the optimal goal point based on this distribution. The accuracy of the map directly affects the plausibility of goal point distribution and, consequently, the accuracy of the predicted target. This makes the model highly dependent on map precision, widening the performance gap between HD and SD maps.

To investigate how the usage of the end-to-end framework influences the gap, we conducted experiments using different map inputs in both end-to-end (our method) and non-end-to-end architectures (original HiVT and DenseTNT models). **We found that the end-to-end approach not only improved overall performance but also reduced the gap between HD and SD map performance.** For the anchor-free model, the gap between minADE and minFDE narrows by 73% (from 0.0130 to 0.0035) and 44% (from 0.0144 to 0.0081). For the anchor-based model, the gap decreases by 87% on minADE and 93% on minFDE. Notably, the narrowed gap does not mean that maps are unnecessary, because an anchor-free model without maps has significantly poorer performance. It states that the precision of maps is less important in end-to-end architecture. This aligns with our hypothesis that under the end-to-end architecture, the motion prediction model directly incorporates visual scene information. It may capture features such as lane markings or road boundaries from BEV features, so the BEV feature partially compensates for the absence of high-precision maps. As a result, maps primarily provide coarse priors, such as road direction and general layout, making map precision less critical to the model's performance.

**Results with predicted SD maps.** We conduct experiments using the predicted SD map generated by MapTRv2 as input for four methods. The results show that when using the predicted SD map, ESDMotion achieves performance comparable to that with ground-truth SD maps. In contrast, using the predicted SD map leads to significant performance degradation for the base and MapUncentaintyPred (Gu et al., 2024a) methods compared to using ground-truth SD maps. For the MapBEVPred (Gu et al., 2024b) method, performance with the predicted SD map is better than using ground-truth SD maps alone but remains significantly lower than when using HD maps. Because

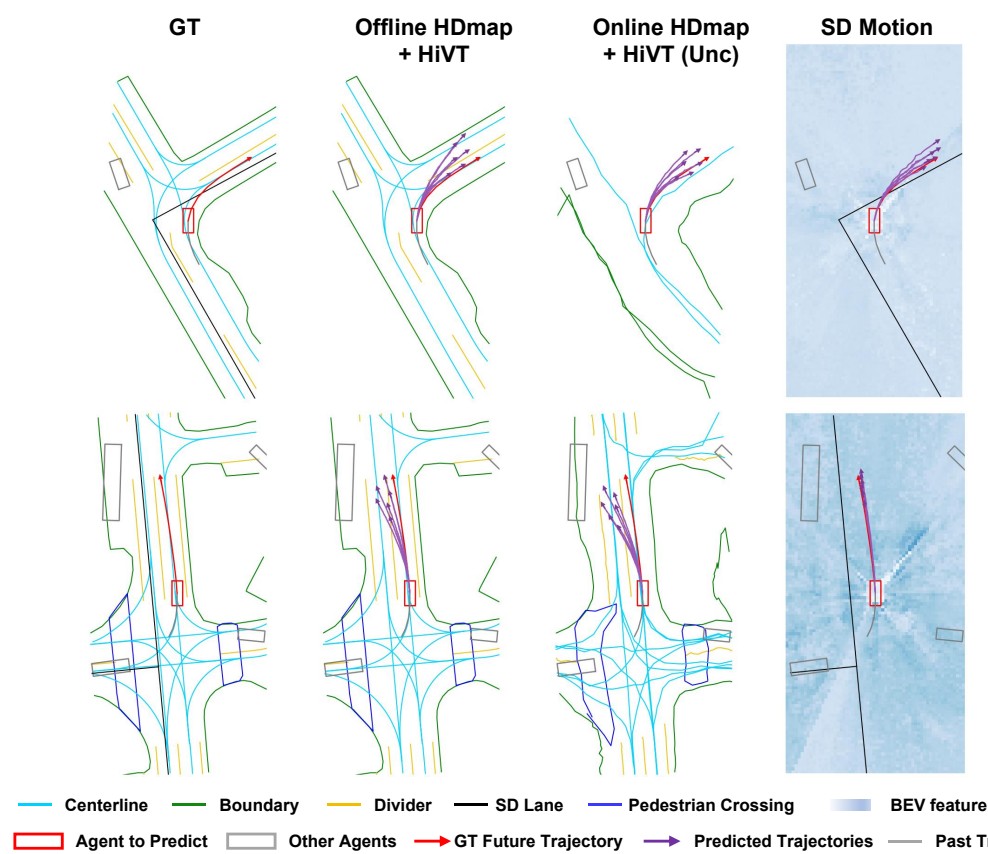

Figure 6: **Qualitative Results.** The purple lines with arrows show six predicted future trajectories, and the red lines represent GT future trajectories. The BEV Feature is colored by the max value of the hidden feature at each grid. Our method provides an accurate prediction of turning and lane following.

Our method is specifically designed for SD maps, it accounts for the relative inaccuracies inherent to SD maps and demonstrates strong robustness. This robustness effectively mitigates the impact of errors introduced by predicted SD maps, ensuring reliable performance under such conditions.

**Qualitative Analysis.** As shown in Fig. 6, we visualize the motion prediction results of our method in two scenarios and compare them with methods that utilize offline or online HD maps. The first row illustrates a scene at a T-junction. The SD map indicates the road direction, signaling turning right, while the BEV feature has stronger responses (darker color) along the turning path. Together, ESDMotion accurately predicts the future motion trajectory. In contrast, the online HD map method underestimates the motion distance, resulting in overly divergent trajectories, some of which encroach into opposite lanes. The error in map estimation leads to inaccurate predictions. The second row depicts a scenario where the vehicle should maintain its current lane after passing through the intersection. The SD map suggests the road direction, while the BEV features highlight the drivable area ahead (indicated by darker colors). ESDMotion correctly predicts that the vehicle should continue straight, whereas the other two methods mistakenly predict a left turn.

### 3.3 ABLATION STUDIES

**Road Observation and Enhanced Road Observation.** Table 4 presents the ablation study results for Road Observation and Enhanced Road Observation on the HiVT-based model. Our Road Observation integrates map features and BEV features and improves motion prediction performance (-0.0052 on minADE and -0.0201 on minFDE). Enhanced Road Observation increases the flexibility of integration, leading to further improvements (-0.0162 on minADE and -0.0418 on minFDE).

Table 3: **Ablation Study on Pseudo Lane Expansion.** We examine several combinations of numbers and distances of parallel lines to expand and choose the best distances as 3 and 6 meters.

| Map | Distances | minADE ↓ | minFDE ↓ | MR ↓ |
|---|---|---|---|---|
| HDMap | - | 0.7166 | 1.2752 | 0.1483 |
| SDMap | - | 1.9735 | 3.8357 | 0.5892 |
| SDMap | [0,3] | 0.9627 | 1.5110 | 0.2451 |
| SDMap | [0,3,6] | 0.7941 | 1.3863 | 0.1627 |
| SDMap | [0,2,4] | 0.8472 | 1.3979 | 0.1722 |
| SDMap | [0,3,6,9] | 0.8132 | 1.3855 | 0.1630 |
| SDMap | Adaptive | 0.7597 | 1.3105 | 0.1523 |

Table 4: **Ablation Study on Enhanced Road Observation.** $F_{norm}$ denotes the normalization of head weights. "-" under "Module" means not using any road observation module.

| Module | $F_{norm}$ | minADE ↓ | minFDE ↓ | MR ↓ |
|---|---|---|---|---|
| - | - | 0.3296 | 0.7080 | 0.0881 |
| Road Obs. | - | 0.3244 | 0.6879 | 0.0832 |
| Enhanced Road Obs. | Softmax | 0.3239 | 0.6918 | 0.0863 |
| Enhanced Road Obs. | Sigmoid | 0.3211 | 0.6879 | 0.0801 |
| Enhanced Road Obs. | None | 0.3134 | 0.6662 | 0.0737 |

Additionally, we tested different normalization methods of head weights. Normalizing weights using either Softmax or Sigmoid functions results in poorer performance, while directly using unrestricted and more expressive weights yields the best performance.

**Pseudo Lane Expansion.** Table 3 presents the ablation study results for Pseudo Lane Expansion on the DenseTNT-based model. The number of expanded parallel lines and their distance from the original SD instance are two critical parameters in Pseudo Lane Expansion. Insufficient or too close expanded lines can limit coverage, potentially missing areas near the vehicle to be predicted, especially when the bias of the SD instance is large. Conversely, an excessive number of lines can lead to interference and increased computational load. Our dynamic strategy adjusts the parameters based on the distribution of SD lanes, enhancing the coverage and accuracy of the generated lines and achieving best performance.

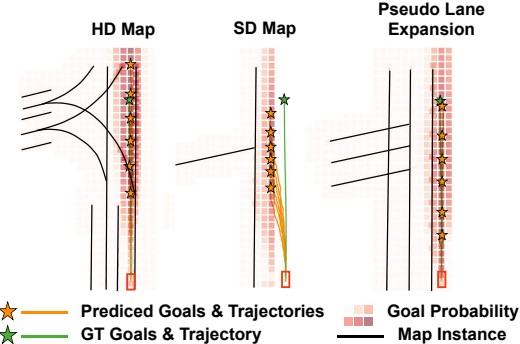

Figure 7: **Visualization of Pseudo Lane Expansion.** Left: the detailed HD map generates appropriate anchors in front of the agent. Middle: the sparser and misaligned SD map results in biased anchors. Right: Pseudo Lane Expansion generates extra anchors near the agent and helps predict the right goal and trajectory.

The visualization illustrates the effect of Pseudo Lane Expansion. In the scenario, the vehicle is on the right side of the road and will drive straight. The provided SD instance correctly indicates the road direction but is located far to the left, resulting in candidate target points that are also positioned on the left side, leading to substantial errors. By applying the Pseudo Lane Expansion method, the SD instance is extended laterally, with the parallel line on the right generating candidate target points ahead of the vehicle, allowing the model to accurately select the target point and predict a trajectory closely aligned with the ground truth.

## 4 CONCLUSION

In this paper, we introduce ESDMotion, a novel end-to-end motion prediction framework that exclusively utilizes SD maps without any HD map supervision. This framework effectively integrates with both anchor-based and anchor-free motion prediction models. To address the low resolution and alignment accuracy of SD maps, we designed two modules called Enhanced Road Observation and Pseudo Lane Expansion. Experiments demonstrate that our model achieves performance comparable to, or even better than online HD mapping-based models. We focused on the performance gap between HD and SD maps, analyzing how the type of motion prediction model and the adoption of end-to-end architectures influence the gap.

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

## A  RELATED WORK

### A.1  HD MAPS AND SD MAPS

**HD Maps.** High-definition(HD) maps contain detailed road information, but their creation requires extensive manual annotation and ongoing maintenance, making them expensive and unscalable (Li et al., 2022a). This leads to the development of various online map estimation methods, which estimate HD maps from camera or LiDAR data. Recent approaches such as MapTR(Liao et al., 2023a; Li et al., 2024b) are mostly based on an encoder-decoder architecture, where BEV (bird's-eye view) features are extracted from sensor data, and a transformer decoder is used to predict various map elements. Moreover, MapEX (Sun et al., 2023) encodes map elements into query tokens and refines the matching algorithm. This results in better performance and robustness. Online map estimation enables autonomous vehicles to operate in areas without Offline HD map coverage, reducing dependency on HD maps. However, these methods still require HD map ground truth for supervised training, making it challenging to obtain sufficient HD map data for large-scale training. Additionally, online HD map estimation consumes computational resources and time.

**SD Maps.** Since obtaining HD maps is costly, easily accessible, and scalable standard-definition (SD) maps garner increasing attention. Currently, many methods utilize SD maps as priors to predict HD maps (Li et al., 2024a). For example, PriorDrive uses a unified vector encoder to effectively encode diverse vector prior maps including SD maps to enhance the robustness and accuracy of online HD map construction. (Zeng et al., 2024) Some studies in this field (Jiang et al., 2024; Zhang et al., 2024) merge SD maps from Open Street Maps (OSM) (Haklay & Weber, 2008) into widely used datasets like nuScenes (Caesar et al., 2019) and OpenLane-V2 (Wang et al., 2023) to make them more available.

## A.2 Motion Prediction with Maps

**Motion prediction with offline HD maps.** Many motion prediction models use offline HD maps to obtain environmental information. Early models typically use rasterized HD maps and encode them with Convolutional Neural Networks (CNNs) (Marchetti et al., 2020; Biktairov et al., 2020; Casas et al., 2020; Gilles et al., 2021). However, high-resolution rasterized maps incur significant storage and computational costs. Recent approaches shift towards vectorized representations of HD maps. In terms of map utilization, some methods such as LaneGCN(Liang et al., 2020), GOHOME (Gilles et al., 2022) and HiVT(Zhou et al., 2022), employ Graph Neural Networks (GNNs) to encode the influence of map elements on vehicle interactions. Other methods like MTR (Shi et al., 2022) and QCNet(Zhou et al., 2023) directly use transformer architectures, leveraging cross-attention mechanisms to fuse map and vehicle features. Some target-based approaches (Zhao et al., 2021; Gu et al., 2021) generate candidate target points based on the map, leading to a stronger dependency on the map. However, most methods rely on HD maps, which are costly to obtain. ViP3D(Gu et al., 2023) combines detection, tracking, and prediction in an end-to-end structure, gets agent information directly from sensor data, but still uses offline HD maps. Currently, there is a lack of motion prediction methods specifically designed for or adapted to SD maps.

**Motion prediction with online HD maps.** Directly inputting online HD maps into motion prediction models is a basic method of online mapping-based motion prediction. However, the error between estimated HD maps and GT HD maps leads to errant behaviors in motion prediction. To address this issue, a highly rated work (Gu et al., 2024a) (**CVPR24 best paper candidate**) extends online map estimation methods to additionally estimate uncertainty, to provide information about potential errors of maps for downstream models. Another study (Gu et al., 2024b) directly uses BEV features generated by online map estimation models as substitutes for vectorized HD maps. In this way, decoding HD maps in upstream models and encoding in the downstream model is omitted, which results in increased speed, decreased information loss, and better performance. However, in this work, the upstream model and downstream model are trained separately and the latter uses stored offline BEV features obtained by the former. This makes the system non-differentiable. Oppositely, in our method, we train all modules in an end-to-end way demonstrating a performance enhancement, and we only use SD maps as map input.

## B Implementation Details

We strictly follow the benchmark protocol in state-of-art online mapping-based motion prediction methods (Gu et al., 2024a;b). For some samples without SD maps in the range, we use an all-zero vector as input. We train ESDMotion on 8 RTX 4090 GPUs with the batch size of 1 on each GPU. We set the learning rate to $2 \times 10^{-4}$ and the number of epochs to 48, with no dropout for faster convergence.

For the BEV encoder, we adopt the official configuration of BEVFormer-base (Li et al., 2022b). The encoder takes a temporal queue of 4 samples as input and obtains BEV features with 6 encoder layers. The BEV feature has a size of 200x200x256 and is in the Lidar coordinate system.

For the motion prediction models, we strictly align the setting of original modules in HiVT and DenseTNT with previous works (Gu et al., 2024a;b) for fair comparison. Specifically, we use a 4-layer temporal transformer, a 1-layer local interaction module, and a 3-layer global interaction module of HiVT. We only add new modules to these two base models without removing existing modules.

## C More details and discussions about DenseTNT and anchor-based model

### C.1 Large sample kernel V.S. Pseudo Lane Expansion

The sample kernel in DenseTNT determines the number and range of dense goal points generated from sparse goals(points on lanes). The larger sample kernel results in the wider coverage of dense

Table 5: **Performance of Two Dense Goals Generation Strategy.**

| Strategy | Kernel Size | minADE↓ | minFDE↓ | MR↓ |
|---|---|---|---|---|
| Original | 2 | 1.9735 | 3.8357 | 0.5892 |
| Original | 6 | 1.2833 | 2.5889 | 0.3451 |
| Original | 10 | 1.3739 | 2.6381 | 0.3890 |
| Pseudo Lane Expansion | 2 | 0.8132 | 1.3855 | 0.1630 |
| Adaptive Pseudo Lane Expansion | 2-3 | 0.7597 | 1.3105 | 0.1523 |

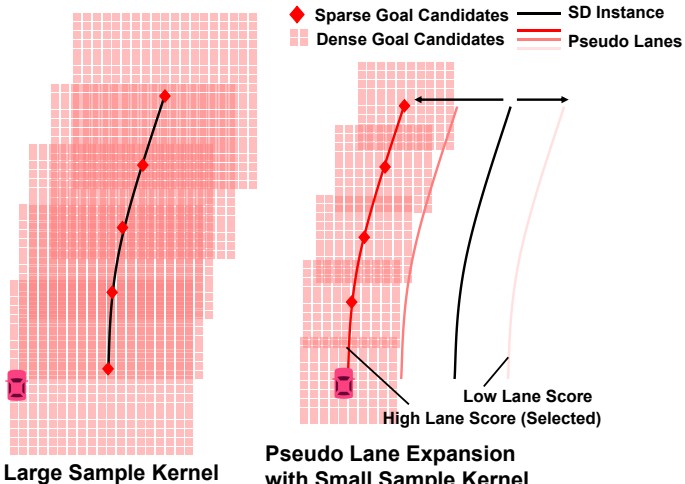

Figure 8: **Large Sample Kernel VS. Pseudo Lane Expansion.**

goal points. However, Pseudo Lane Expansion achieves this in a more efficient and interpretable way.

In DenseTNT, goal point selection is a hierarchical process. The lane scoring is performed first, where features are extracted for each lane of the vectorized map, and scores are computed to select a fixed number of lanes. Then, dense goal points are generated around the points of selected lanes with the sampling kernel.

Simply increasing the sampling kernel size to achieve a coverage similar to pseudo-lane expansion would require a kernel size of nearly 10. This would lead to an excessively large number(more than 2000 for a lane) of densely sampled points, significantly reducing model efficiency, which is shown in Fig. 8 (left). In contrast, Pseudo Lane Expansion uses a smaller sample kernel and generates pseudo lanes that hypothesize the approximate locations of potential drivable paths, as shown in Fig. 8 (right). Through lane scoring, the model identifies the pseudo lanes most "real" and most likely to represent road structures. Dense sampling is then applied only around these selected pseudo-lanes. This approach reduces the number of densely sampled points and is specifically designed to adapt to the coarser and less aligned SD maps. The Tab. 5 shows that our Pseudo Lane Expansion achieves better performance than simply increasing the size of the sample kernel.

### C.2 PSEUDO LANES WITH EMPTY MAPS

Although SD maps are globally covered, there are small areas like parking lots that are not covered. In fact, this is a common limitation of methods that rely entirely on maps for goal point selection. We make adaptations to DenseTNT to handle such special scenarios. As shown in Fig. 9, We defined a set of lanes at specific angles to simulate the potential driving directions of the vehicle. With the positions of other vehicles and visual information provided by the BEV feature, the model selects appropriate goal points and paths in these situations.

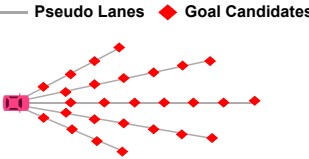

Figure 9: **Pseudo Lanes with Empty Maps.**

Table 6: **Protocols and focus of works in end-to-end motion prediction.**

| Methods | Map information | Agents' Information | Focus |
|---|---|---|---|
| ViP3D (Gu et al., 2023) | GT HD maps | Detection results | The cooperative relations of detection and motion prediction |
| PiP (Jiang et al., 2022) | Predicted HD maps | Detection results | The interaction between detection and online mapping |
| ESDMotion (Ours) | GT/predicted SD maps | GT | Replacing HD Map with low-cost SD Map for motion prediction |

# D PROTOCOLS AND FOCUS OF END-TO-END WORKS IN MOTION PREDICTION

With the rapid development of end-to-end motion prediction, many valuable works emerge with different protocols and focus in the field of motion prediction, as shown in Tab. 6.

ViP3D (Gu et al., 2023) is the first fully differentiable vision-based approach to predict future trajectories of agents. For the prediction module in ViP3D, the input information of agents is obtained from the detection and tracking module and the input maps are GT HD maps. This setting is suitable for studying the cooperative relations of detection and motion prediction.

PiP (Jiang et al., 2022) is the first end-to-end Transformer-based framework that jointly and interactively performs online mapping, object detection, and motion prediction. For the prediction module in PiP, the input information of agents is from the perception model, and the input map is estimated online. This protocol is designed to explore the interaction between detection and online mapping, and its influence on downstream motion prediction task.

ESDMotion focuses on replacing HD Map with low-cost SD Map for motion prediction. End-to-end feature usage is our solution for the issues caused by using SD maps. To study the influence of map in a decoupled way, we adopt the protocols of MapUncertainty (Gu et al., 2024a) and BEVPred (Gu et al., 2024b). We use ground-truth information of agents and SD maps as input. With this protocol, the conclusion would not be influenced by the detection module.

# E FUTURE WORK

There are two important aspects worth further investigation regarding the usage of SD maps. (1) Investigating the performance of motion prediction using SD maps in complex scenarios. Two insightful studies (Weng et al., 2024; Li et al., 2024c) have highlighted that most road structures in the nuScenes dataset are relatively simple. It is valuable to explore the differences between utilizing HD maps and SD maps in complex road structures, such as intricate intersections and detailed ramps. (2) Exploring methods for utilizing SD maps in other downstream tasks such as planning.

