# OpenReview forum: "ESDMotion: End-to-end Motion Prediction Only with SD Maps"
_ICLR.cc/2025/Conference — Submitted to ICLR 2025_

### Official Review · Reviewer_9KQE · 2024-10-30

**Soundness:** 2
**Presentation:** 2
**Contribution:** 2
**Rating:** 5
**Confidence:** 4

**Summary:**

The paper presents a method to integrate Standard Definition (SD)-maps information in trajectory prediction models. The model jointly integrates sensor-based features (BEV maps) with SD-maps features with a deformable attention. Additionally, to account from the poor spatial localization of SD-maps, and missing fine-grained details, the method suggests a simple “pseudo-lane expansion” where artificial lanes are created parallelly to the main lane of the SD map. Each of these lanes serves in the deformable attention (“Enhanced Road Observation”) and provides goal candidates in anchor-based methods. Experiments are conducted on nuScenes.

**Strengths:**

* Introducing SD-maps, that are readily available with cheap and crowd-sourced maintenance, in trajectory prediction models is a good idea. While some previous works exist to use SD maps as priors to build HD maps, the impact on downstream application (here trajectory prediction) is an underexplored and important topic.

* The proposed modules are motivated, and their instantiation is sound.

* The method is designed for both anchor-based and anchor-free methods.

* The paper is well illustrated.

* The discussion on the contribution differences between SD / HD maps in anchor-based/anchor-free models was appreciated.

**Weaknesses:**

`[Model presentation]` The presentation of the method (section 3) could be significantly improved. In the current form, the model presentation interleaves the general trajectory prediction framework (from previous works), contributions of the work (enhanced road observation, pseudo lane expansion), high-level insights, and technical details (e.g. instantiation of the encoders, etc…). The presentation of the instantiation of the method for either anchor-based or anchor-free models can be also improved. For instance, the title of Section 3.2 targets anchor-free models but “the anchor-based model DenseTNT” is discussed at the end.

`[Pseudo-lane expansion]` DenseTNT already “expands” the lanes by densely sampling candidate goals around the lanes. It looks like the proposed pseudo-lane expansion simply amounts to increasing the size of the sampling kernel in denseTNT. Can the authors comment on this?

`[Prior works]`
There are prior works that try to infer HD online maps from SD-maps and sensor-data [1,2,3]
The reviewer suggests discussing these approaches in the related work section, as it offers another way to integrate SD maps.

* [1] Mind the map! Accounting for existing maps when estimating online HDMaps from sensors. Sun et al. 2023
* [2] Driving with Prior Maps: Unified Vector Prior Encoding for Autonomous Vehicle Mapping, Zeng et al. 2024
* [3] Local map Construction Methods with SD map: A Novel Survey, Li et al. 2024

`[Fair baselines]` The comparison between ESDMotion and “Unc”/”BEVPred” is unfair as ESDMotion uses SD maps which the latter ones do not (lines 369-374). Related to the previous remark, it would be valuable to compare ESDMotion with “Unc” + “online HD mapping based on SD maps” and “BEVPred” + “online HD mapping based on SD maps”.

`[Impact of the Enhanced Road Observation]` The improvement brought by the Enhanced Road Observation module is somewhat limited (Table 4).

`[Map updates]` The review agrees that HD maps must be often updated (l.43). However, I believe the use of SD maps does not solve this problem as SD maps should be updated as well.

`[Multimodal future prediction]` How many modes are generated by the models? (l.362,363).

`[Title]` The use of the word “only” in the title is questionable. There are some works doing end-to-end motion prediction without any maps (SD / HD).

`[Writing clarity]` The many paper typos, syntax and grammar issues hurt the readability of the paper. A thorough proof-read is strongly recommended. A non-exhaustive list of typos are shown below.
* “ground truth” → Hyphen is needed when used as an adjective
* Multiple spaces “ “ are missing, e.g., before citations, parenthesis, after “.” etc…
* Line 220. Is it Sec 4.3?
* Line 334. The subject is missing. “We”?
* Line 413 “tab1”
* Line 296 “an”
* Please refer to Figure 1 and Figure 6 in the text where appropriate.
* Figure 3 is discussed after Figure 4.
* Some notations are introduced but never used (e.g., l.293,335)
* Opening double quote signs are off
* Illustrations are pixelated, pdf figures are generally preferred as they also allow for text selection and search.
* Tab 3 and 4 should be at the top of the page.
* In the reviewer’s opinion, the usage of bold sentences is too excessive.

**Questions:**

Beyond the need for improved paper presentation and proofreading, answers to the points on `[Pseudo-lane expansion]` and `[Fair baselines]` may lead to a reconsideration of the reviewer’s recommendation.

---

> ### Author Response · Authors · 2024-11-21
>
> >**Q1. [Model presentation] The presentation of the method (section 3) could be significantly improved.**
>
> Thank you for your suggestion. We re-write the method section to make it clear and hierarchical.
> - In Section 2.1, we present the overall structure of ESDMotion (2.1.1) and the modules required to extend the original motion prediction model into SD map oriented end-to-end framework (2.1.2). We indicate which modules are from existing work and which are proposed by us.
> -  In Section 2.2, we discuss the challenges faced by anchor-free models when using SD maps as input, and introduce the corresponding solution, the Enhanced Road Observation module.
> - In Section 2.3, we address the challenges and solutions for anchor-based models, specifically the Pseudo-Lane Expansion and the newly proposed Adaptive Pseudo Lane Expansion.
>
> Please check [our updated paper](https://openreview.net/pdf?id=sEJYPiVEt4) and we sincerely thanks for your detailed suggestion.
>
> >**Q2. [Pseudo-lane expansion] It looks like the proposed pseudo-lane expansion simply amounts to increasing the size of the sampling kernel in denseTNT.**
>
> Thanks for your in-depth thinking. We clarify the difference below.
>
> Simply increasing the sampling kernel size to achieve a coverage similar to pseudo-lane expansion would require a kernel size of nearly 10. This would lead to an excessively large number (2000+ for a lane) of densely sampled points, significantly reducing model efficiency.
>
> In contrast, Pseudo Lane Expansion uses smaller sample kernel and generates pseudo lanes that hypothesizes the approximate locations of potential drivable paths. Through lane scoring, the model identifies the pseudo lanes most “real” and most likely to represent road structures. Dense sampling is then applied only around these selected pseudo-lanes. This approach reduces the number of densely sampled points and is specifically designed to adapt to the coarser and less aligned SD maps. It improves efficiency and offers greater interpretability. There is a [figure](https://postimg.cc/3dxW2Khk) showing the difference. The Table below shows that our Pseudo Lane Expansion achieves better performance than simply increasing the size of the sample kernel.
>
> | Strategy              | Kernel Size | minADE $\downarrow$ | minFDE $\downarrow$ | MR $\downarrow$ |
> | --------------------- | ----------- | ------------------- | ------------------- | --------------- |
> | Original              | 2           | 1.9735              | 3.8357              | 0.5892          |
> | Original              | 6           | 1.2833              | 2.5889              | 0.3451          |
> | Original              | 10          | 1.3739              | 2.6381              | 0.3890          |
> | Pseudo Lane Expansion | 2           | 0.7941              | 1.3863              | 0.1627          |
>
> We add the above discussions and experiments to Appendix C.1 of [our updated paper](https://openreview.net/pdf?id=sEJYPiVEt4) and thanks for raising your advice to better illustrate the advantages of ESDMotion.
>
> >**Q3 [Prior works]Add prior works that try to infer HD online maps from SD-maps and sensor-data.**
>
> Indeed, these are pioneering and inspiring works for ESDMotion. We add more discussions about them to [our updated paper](https://openreview.net/pdf?id=sEJYPiVEt4) and thanks for your advice to better position this work.

---

> > ### Author Response · Authors · 2024-11-21
> >
> > >**Q4. [Fair baselines] The comparison between ESDMotion and “Unc”/”BEVPred” is unfair as ESDMotion uses SD maps which the latter ones do not.**
> >
> > Thank you for your suggestion. For fair comparison, we addtionally provide GT SD maps for the “Base”, “Unc”, and “BEVPred” methods.
> >
> > Results with HiVT:
> > | Methods     | Maps                   | minADE $\downarrow$ | minFDE $\downarrow$ | MR $\downarrow$ |
> > | ----------- | ---------------------- | ------------------- | ------------------- | --------------- |
> > | Base        | MapTRv2-CL + GT SD map | 0.3521              | 0.7311              | 0.0711          |
> > | Unc         | MapTRv2-CL + GT SD map | 0.3562              | 0.7209              | 0.0646          |
> > | BEVPred     | MapTRv2-CL + GT SD map | 0.3641              | 0.7374              | 0.0723          |
> > | ESDMotion++ | GT SD map              | 0.3114              | 0.6692              | 0.0727          |
> >
> > Results with DenseTNT:
> > | Methods     | Maps                   | minADE $\downarrow$ | minFDE $\downarrow$ | MR $\downarrow$ |
> > | ----------- | ---------------------- | ------------------- | ------------------- | --------------- |
> > | Base        | MapTRv2-CL + GT SD map | 0.7534              | 1.2983              | 0.1505          |
> > | Unc         | MapTRv2-CL + GT SD map | 0.7629              | 1.3185              | 0.1529          |
> > | BEVPred     | MapTRv2-CL + GT SD map | 0.7557              | 1.3276              | 0.1562          |
> > | ESDMotion++ | GT SD map              | 0.7597              | 1.3105              | 0.1523          |
> >
> > Additionally, we include experiments where predicted HD maps are used as inputs to ESDMotion, as well as experiments using SD maps predicted by MapTRv2 as inputs for all methods. Please check Table 1 of [our updated paper](https://openreview.net/pdf?id=sEJYPiVEt4). These results provide further fair comparisons under equivalent input conditions.
> >
> > **Across all input settings (predicted HD maps + GT SD maps, predicted SD maps, GT HD maps, and predicted HD maps), our method outperforms both Unc and BEVPred**.
> >
> > We add the above discussions and experiments to Table 1 of [our updated paper](https://openreview.net/pdf?id=sEJYPiVEt4) and thanks for raising your advice to better illustrate the advantages of ESDMotion.
> >
> >
> > >**Q5. [Impact of the Enhanced Road Observation] The improvement brought by the Enhanced Road Observation module is somewhat limited (Table 4).**
> >
> > The Enhanced Road Observation module reduces minADE by 5%. In fact, this improvement is similar to the improvements achieved by previous methods such as Unc (4% on minADE).
> >
> > >**Q6. [Map updates]The review agrees that HD maps must be often updated (l.43). However, I believe the use of SD maps does not solve this problem as SD maps should be updated as well.**
> >
> > We agree that SD maps also need to be updated due to urban development and road construction. However, the cost of updating SD maps is significantly lower compared to HD maps, as SD maps only include simple road skeleton information, as currently maintained by servers like Google Map. We have revised the phrasing in the paper to clarify this point and avoid misunderstanding.
> >
> >
> > >**Q7. [Multimodal future prediction] How many modes are generated by the models?**
> >
> > Following the protocol in Unc/BEVPred, all the models predict 6 modes of future trajectories.
> >
> > >**Q8. [Title] The use of the word “only” in the title is questionable. There are some works doing end-to-end motion prediction without any maps (SD / HD).**
> >
> > Thanks for your suggestion. To avoid misunderstanding, we change our title to **ESDMotion: SD Map Oriented Motion Prediction Enhanced by End-to-end Learning**. We revise all relevant statements in [our updated paper](https://openreview.net/pdf?id=sEJYPiVEt4) and describe ESDMotion as a motion prediction method that aims to replace HD map usage with SD map. This phrasing is more precise and clearly distinguishes our approach from existing methods.
> >
> > >**Q9. [Writing clarity] The many paper typos, syntax and grammar issues hurt the readability of the paper.**
> >
> > Thanks for your suggestion. We fixed the typos, figures and syntax and grammar issues. Please check [our updated paper](https://openreview.net/pdf?id=sEJYPiVEt4).

---

> > > ### Author Response · Authors · 2024-11-25
> > >
> > > Dear Reviewer,
> > >
> > > We hope our response regarding **Pseudo-lane expansion** and **Fair baselines** solves your concerns. If there is any further question, we are glad to discuss.
> > >
> > > Since the discussion period comes to end, we humbly request you to consider improving the score if you agree that the quality of the updated manuscript improves and satisfies the stanard of positive scores.
> > >
> > > Thanks,
> > >
> > > Authors of Submission 2920

---

> > > ### Comment · Reviewer_9KQE · 2024-11-27
> > >
> > > The reviewer thanks the authors for the thorough answer and paper revision.
> > >
> > > Q2. It is now clear.
> > >
> > > Q4. The reviewer is not sure to understand which line is  “Unc” + “online HD mapping based on SD maps” and “BEVPred” + “online HD mapping based on SD maps” in the new experiments.
> > > The question implied the following experiments:
> > > * Step 1: Train a model “A” that takes (GT SD-inputs)+(camera images) as inputs and learns to predict HD-map. Model “A” may be any model suggested in Q3.
> > > * Step 2: Use the predicted HD-maps from model A as input for Unc or BEVPred.
> > > Do the new results include this experiment?
> > >
> > > Moreover, the reviewer does not understand the meaning of “MapTRv2-CL + GT SD map”. Does it mean that there are two maps: a predicted HD-maps from MapTRv2-CL and the GT SD maps that are concatenated (?) together and used as the input map?
> > >
> > > Thanks for correcting typos and reformatting section 3.
> > > Still, the reviewer insists again that the paper undergoes meticulous proofreading.
> > > “MapUncentaintyPred” (L.430)
> > > “, We” (L.209)
> > > “, We” (L.807)
> > > …

---

> ### Author Response · Authors · 2024-11-27
>
> Thanks for your reply!  We response below:
>
> - Regarding Q4, your proposed two-step setting could be one way to make the comparison more fair.  However,  **[1] and [2] are not open sourced yet while [3] is a survey**.  [1] does not give any link in their paper (https://arxiv.org/pdf/2311.10517) and we could not find a github repo after searching while [2] is an empty repo yet (https://github.com/missTL/PriorDrive).
> - Indeed, your understanding is correct. We find a substitute way to solve your concern regarding the fairness: *There are two maps: a predicted HD-maps from MapTRv2-CL and the GT SD maps that are concatenated together and used as the input map*.
> - In this way, these methods have the SD map as inputs as well. In our humble opinion, **it is fair since all methods have SD maps when conducting motion prediction** while online mapping based methods have the extra advantages of HD map supervision.  We are wondering about your opinion regarding this setting.
>
> Regarding the typos, we will revise them.
>
> We want to express our sincere thanks for your in-depth reading and thinking of the paper. Your commet helps us dig deeper into the new setting and demonstrate more comprehensive comparisons with existing works, especially on the points of fair comparison and relations with DenseTNT under the unexplored sparse SD map setting. Additionally, your careful writing advice also helps us a lot, which leads to a more clear Method section.
>
> As the first work to explore in this direction, we humply request you to consider improving the score if our response solves your concerns.  We believe our findings and solutions could be insightful for the future works in the field.

---

> > ### Author Response · Authors · 2024-11-29
> >
> > Dear Reviewer,
> >
> > As the last day that reviewers may post a message is December 2nd, we are wondering whether your remaining concerns are solved and we humbly request a positive score. If there are any unclear points, we are eager to discuss with you.
> >
> > Thanks
> >
> > Authors of Submission 2920

---

> > > ### Author Response · Authors · 2024-12-02
> > >
> > > Dear Reviewer,
> > >
> > > **Since we have responded your existing concerns with solid experiments and validations and there is no new concern raised**, we humbly request you to consider adjusting scores accordingly. If there are any further concerns, we are glad to respond.
> > >
> > > Thanks
> > >
> > > Authors of Submission 2920

---

> > > ### Comment · Reviewer_9KQE · 2024-12-02
> > >
> > > The reviewer appreciates the authors' efforts to clarify certain aspects of the paper and to include fair baselines.
> > >
> > > While the recommended rating has been raised, it still falls short of the acceptance threshold. This is due to the technical contributions remaining limited, both in terms of methodological innovations and the empirical results presented.

---

> > > > ### Author Response · Authors · 2024-12-02
> > > >
> > > > Thanks for your reply and improve the score. We totally respect your opinion.
> > > >
> > > > However,  we respectfully could not agree with your two final evidence:
> > > >
> > > > From *methodological innovations* perspective, we are the **first** to replace HD Map with SD Map for motion prediction and the proposed modules improve the minADE of vallina replacement from 0.3998  to 0.3134 for HiVT and from 1.2117 to 0.7597 for DenseTNT.
> > > >
> > > > From *empirical results* perspective, we improve minADE from 0.3904 to 0.3134 for HiVT and from 0.7630 to 0.7597 for DenseTNT, compared to **BEVPred (ECCV 2024),  under the disadvantages of SD Map**!
> > > >
> > > > Based on [ICLR 2025 reviewer guideline](https://iclr.cc/Conferences/2025/ReviewerGuide) "Review for a Paper where Leaning-to-Reject",   we argue that **more specific reasons should be given for reject rating.** For example,  which work has made our work not novel enough and which metrics make our emperical results not strong enough?
> > > >
> > > > Thanks
> > > >
> > > > Authors of Submission 2920

---

### Official Review · Reviewer_TmWe · 2024-11-02

**Soundness:** 2
**Presentation:** 3
**Contribution:** 2
**Rating:** 5
**Confidence:** 4

**Summary:**

To address the dependency of trajectory prediction models on high-cost offline high-precision maps or online mapping modules, this paper proposes ESDMotion. ESDMotion uses surround-view images and globally covered SD maps as environmental input for trajectory prediction, and experiments were conducted on nuScenes. The method proposed in this paper to use SDMap is interesting. However, I have concerns about the contributions of this paper and the fairness of some experiments.

**Strengths:**

1.Direct and clear motivation with coherent writing.
2.The approaches of better utilizing SDMap (Pseudo Lane Expansion and Enhanced Road Observation) are interesting.

**Weaknesses:**

1.The paper is misleading (including the title and contribution 1) readers. It does not only rely on SD Map for trajectory prediction but rather combines surround-view images. The surround-view images provide crucial local environmental information, which models like HiVT and DenseTNT do not utilize.
2.The emphasis on the end-to-end architecture of ESDMotion as a contribution is questionable. With BEVPred and ViP3D's work[1-2] and open-source code as a foundation, training a model end-to-end from images to BEV features to trajectory prediction presents no real challenges or contributions. The contribution of this paper seems lacking. Although the code[3] provided by BEVPred replaces mapping module features with HD maps as input for trajectory prediction, it can easily be modified to enable end-to-end training of the BEV encoder and trajectory prediction model.
3.From Table 2, it can be seen that for HiVT, the minADE and minFDE using HDMap and SDMap are quite similar. While ESDMotion claims to improve minADE and minFDE through an end-to-end architecture or BEV features, as I mentioned in point 2, I believe this end-to-end architecture should not be counted as a contribution of this paper. This improvement can also be observed in the comparison between BEVPred and offline HDMap in Table 1.


References:
[1]. Gu X, Song G, Gilitschenski I, et al. Accelerating online mapping and behavior prediction via direct bev feature attention[J]. arXiv preprint arXiv:2407.06683, 2024.
[2]. Gu J, Hu C, Zhang T, et al. Vip3d: End-to-end visual trajectory prediction via 3d agent queries[C]//Proceedings of the IEEE/CVF Conference on Computer Vision and Pattern Recognition. 2023: 5496-5506.
[3]. https://github.com/alfredgu001324/MapBEVPrediction

**Questions:**

1.Although SD maps are globally covered, there are small areas like parking lots (which indeed exist in nuScenes) that are not covered. How does DenseTNT handle such situations?
2.Why is ESDMotion unable to outperform BEVPred on DenseTNT?

---

> ### Author Response · Authors · 2024-11-21
>
> Thanks for your acknowledgement and kind advice. Regarding your concerns, we give responses below:
>
>
> >**Q1. The statement of “only use SD maps” is misleading readers.**
>
> Thanks for your suggestion. We change our title to **ESDMotion: SD Map Oriented Motion Prediction Enhanced by End-to-end Learning**. We revise all relevant statements in [our updated paper](https://openreview.net/pdf?id=sEJYPiVEt4) and describe ESDMotion as a motion prediction method that “utilizes only SD map for map information”. This phrasing is more precise and clearly distinguishes our approach from methods that leverage both SD map priors and HD map supervision. Any further advice is welcomed.
>
> >**Q2. The emphasis on the end-to-end architecture of ESDMotion as a contribution is questionable.**
>
> We agree that with the rapid development in recent years, end-to-end learning in autonomous driving becomes more and more common. Our method is certainly not the first end-to-end approach.
>
> However, our contribution lies in that:
> - We aim to address a previously unexplored problem: using SD maps as the sole map information for motion prediction.
> -  We find out issues (low resolution and poor alignment) caused by using SD maps. To overcome these issues, **we design an end-to-end framework and tailored modules to better utilize SD maps**.
> -  As a result, our method demonstrates that even with the low-cost SD maps, it is possible to achieve motion prediction performance comparable to that using HD maps
> - In fact, our approach is the first to specifically design a method for SD maps in the context of motion prediction and the first end-to-end approach addressing this problem. This focus on SD maps as a practical, cost-effective alternative represents the core of our contribution.
>
> We compare the contributions of the three works below:
>
> | Methods                                     | Map Usage           | Main Idea (Contribution)                                                                                            |
> | ------------------------------------------- | ------------------- | ------------------------------------------------------------------------------------------------------------------- |
> | [ViP3d](https://arxiv.org/pdf/2208.01582)   | HD map(Input)       | The first work to leverage fine-grained visual information to motion prediction through end-to-end learning .       |
> | [BEVPred](https://arxiv.org/pdf/2407.06683) | HD map(Supervision) | Accelerate online mapping based motion prediction through a structure similiar to end-to-end learning.              |
> | ESDMotion                                   | SD map(Input)       | The first work designed for SD maps on motion prediction with minimum performance drop through end-to-end learning. |
>
> >**Q3. From Table 2, it can be seen that for HiVT, the minADE and minFDE using HDMap and SDMap are quite similar. While ESDMotion claims to improve minADE and minFDE through an end-to-end architecture or BEV features, as I mentioned in point 2, I believe this end-to-end architecture should not be counted as a contribution of this paper. This improvement can also be observed in the comparison between BEVPred and offline HDMap in Table 1.**
>
> As an anchor-free approach, the precision of the map (SD and HD) has a relatively smaller impact on HiVT comapred to anchor-based DenseTNT. In fact, for the original HiVT model, there is still a measurable gap in performance between using GT HD map and GT SD map (0.013 in minADE). Our method reduces this gap (to 0.0035 in minADE). As mentioned in response to point 2, **the primary goal of our work is to leverage SD maps rather than merely building an end-to-end model**. The end-to-end part is to fix the issue caused by SD maps. Our method demonstrates that by using more accessible SD maps, we can achieve performance that surpasses current methods relying on HD maps. **We believe the SD map oriented designs constitutes a meaningful contribution of this paper**.

---

> ### Author Response · Authors · 2024-11-21
>
> >**Q4. Although SD maps are globally covered, there are small areas like parking lots (which indeed exist in nuScenes) that are not covered. How does DenseTNT handle such situations?**
>
> **This is indeed a common limitation of anchor based methods like DenseTNT which we believe should not be the reason to punish the ESDMotion**.
>
> Additionally, inspired by your concerns, we try to make adaptations to DenseTNT to handle such special scenarios. In areas not covered by maps, we define a set of pseudo lanes at specific angles to simulate the potential driving directions of the vehicle, as shown in this [figure](https://postimg.cc/ctSKrp1c). With the positions of other vehicles and visual information provided by the BEV feature, the model could select appropriate goal points and paths in these situations.
>
> The DenseTNT adapted to empty-map situation is named DenseTNT+, and the ESDMotion with this adaption is ESDMotion+. This adaption improves performance.
>
>
> | Models     | Maps   | minADE $\downarrow$ | minFDE $\downarrow$ | MR $\downarrow$ |
> | ---------- | ------ | ------------------- | ------------------- | --------------- |
> | DenseTNT   | SD map | 1.2117              | 1.9849              | 0.2776          |
> | DenseTNT+  | SD map | 1.1746              | 1.9421              | 0.2630          |
> | ESDMotion  | SD map | 0.7941              | 1.3863              | 0.1627          |
> | ESDMotion+ | SD map | 0.7783              | 1.3784              | 0.1616          |
>
> We add the above discussions and experiments to Appendix C.1 of [our updated paper](https://openreview.net/pdf?id=sEJYPiVEt4) and thanks for expressing your concerns to improve the method.
>
>
> >**Q5. Why is ESDMotion unable to outperform BEVPred on DenseTNT?**
>
>
> The reason is that: **Anchor-based methods like DenseTNT use lanes to generate goal candidates**. The sparesity and misalignment of lanes in SD map would lead to missing of nearby candidates and thus bad performance. As a result, compared to anchor-free methods like HiVT, the gains of DenseTNT is undesirable.
>
> Inspired by you and other reviewers' thoughts, we improve DenseTNT under SD map scenario in two ways:
> 1. There are small areas like parking lots that are not covered by maps. For these areas, we define a set of pseudo lanes at specific angles capturing the potential driving directions to generate goal candidates, as shown in this [figure](https://postimg.cc/hhgXPL3b). This updated version is named DenseTNT+ and ESDMotion+. As shown in Table below, such modifications explicitly improve the performance of DenseTNT.
>
> | Models    | Maps      | minADE $\downarrow$ | minFDE $\downarrow$ | MR $\downarrow$ |
> | ----------| ---       | ------------------- | ------------------- | --------------- |
> | DenseTNT  | SD map | 1.2117              | 1.9849              | 0.2776          |
> | DenseTNT+ | SD map | 1.1746              | 1.9421              | 0.2630          |
> | ESDMotion | SD map | 0.7941              | 1.3863              | 0.1627          |
> | ESDMotion+| SD map | 0.7783              | 1.3784              | 0.1616          |
>
> 2. In the origin version of Pseudo Lane Expansion, we expand a fixed number of parallel lines at uniform distances, which introduces too many candidates and thus hurts performance. To reduce those unnecessary candidates, **we propose to dynamically adjust the parameters of Pseudo Lane Expansion based on the position and density of SD map.** Specifically, for misaligned SD map as shown in [figure](https://postimg.cc/qNZJsnLY) (upper part), we generate more pseudo lanes on the side closer to the vehicle and reduce the number on the opposite side. To deal with different density of SD lanes as shown in [figure](https://postimg.cc/qNZJsnLY) (lower part)/Figure 5 in our paper, we increase the number of pseudo lanes to ensure better coverage for spare ones while decrease the number for dense ones. This further updated version is named ESDMotion++. As shown below, the performance is further enhanced, which
>
> | Models      | Maps   | minADE $\downarrow$ | minFDE $\downarrow$ | MR $\downarrow$ |
> | ----------- | ------ | ------------------- | ------------------- | --------------- |
> | DenseTNT+BEVPred | HD map | 0.7630              | 1.3609              | 0.1576          |
> | ESDMotion   | SD map | 0.7941              | 1.3863              | 0.1627          |
> | ESDMotion+  | SD map | 0.7783              | 1.3784              | 0.1616          |
> | ESDMotion++ | SD map | 0.7597              | 1.3105              | 0.1523          |
>
> We add these discussions, methods, and experiments into [our updated paper](https://openreview.net/pdf?id=sEJYPiVEt4). We sincerely thank you for the constructive comments.

---

> > ### Author Response · Authors · 2024-11-25
> >
> > Dear Reviewer,
> >
> > We hope our response regarding **Anchor-based Method with SD Map** solves your concerns. If there is any further question, we are glad to discuss.
> >
> > Since the discussion period comes to end, regarding the novelty and contributions of this work, if you have any suggestions or concerns regarding **the major motivation of SD map oriented motion prediction**, please do not hesitate to discuss.
> >
> > We humbly request you to consider improving the score If you agree that the quality of the updated manuscript improves and satisfies the stanard of positive scores.
> >
> > Thanks,
> >
> > Authors of Submission 2920

---

> > > ### Author Response · Authors · 2024-11-29
> > >
> > > Dear Reviewer,
> > >
> > > As the last day that reviewers may post a message is December 2nd, we are wondering about your opinion and whether our responses solve your concerns. We humbly request you to consider a positive score if there is no more concerns and we are glad to solve any further concerns.
> > >
> > > Thanks
> > >
> > > Authors of Submission 2920

---

> > > > ### Author Response · Authors · 2024-12-02
> > > >
> > > > Dear Reviewer,
> > > >
> > > > **Since we have responded your existing concerns with solid experiments and validations and there is no new concern raised**, we humbly request you to consider adjusting scores accordingly. If there are any further concerns, we are glad to respond.
> > > >
> > > > Thanks
> > > >
> > > > Authors of Submission 2920

---

### Official Review · Reviewer_Arah · 2024-11-04

**Soundness:** 3
**Presentation:** 3
**Contribution:** 2
**Rating:** 6
**Confidence:** 3

**Summary:**

The paper presents ESDMotion, an end-to-end motion prediction framework that uses only standard-definition (SD) maps for autonomous driving motion prediction to address the limitations of high-definition maps. It integrates BEV features from raw sensor data into existing motion prediction models and presents Enhanced Road Observation and Pseudo Lane Expansion to handle the challenges brought by SD maps. Experiments on the nuScenes dataset show competitive performance compared to online mapping-based methods and mitigate the performance gap between HD and SD maps.

**Strengths:**

1. This paper proposes an end-to-end motion prediction framework that relies solely on SD maps, addressing the scalability and cost issues associated with HD maps.
2. This paper introduces Enhanced Road Observation and Pseudo Lane Expansion to enhance feature fusion and anchor generation for SD maps, addressing the limitations of using SD maps in motion prediction.
3. The proposed ESDMotion obtains comparable or even better performance compared to methods based on HD maps or GT maps.

**Weaknesses:**

1. Although the proposed approach ESDMotion obtains improvements in Tab.1&2, the baseline methods (DenseTNT and HiVT) are too old. It will be better to adopt the latest methods.
2. The proposed ESDMotion aims for end-to-end motion prediction however it still relies on SD maps as extra inputs, which is different from previous end-to-end works[1,2].
3. The use of SDMap feature fusion as a way to enhance BEV features is already common; the authors focus on fusing BEV features and SDMap prior features around the agent, which is innovative but lacks sufficient novelty.
4. The nuScenes dataset mostly contains simple road structure scenarios (PARA-Drive, BEVPlanner), in some complex scenarios, will the ESDMotion scheme proposed by the authors perform more excellently, for example, compared with PARA-Drive?


References\
[1] Jiang et al. Perceive, Interact, Predict: Learning Dynamic and Static Clues for End-to-End Motion Prediction. arXiv 2022.\
[2] Gu et al. ViP3D: End-to-end Visual Trajectory Prediction via 3D Agent Queries. CVPR 2023.

**Questions:**

N/A

---

> ### Author Response · Authors · 2024-11-21
>
> Thanks for your acknowledgement and kind advice. Regarding your concerns, we give responses below:
>
> >**Q1. The baseline methods (DenseTNT and HiVT) are too old.**
>
> We agree that there are more advanced methods emerged recently. We choose DenseTNT and HiVT as baselines primarily for fair comparison with existing online map based motion prediction methods (MapUncertaintyPrediction and MapBEVPrediction). According to your kind advice, we implement a more recent motion prediction model [MTR](https://arxiv.org/abs/2209.13508), into the ESDMotion framework. Due to time constraints, we complete four pioneering experiments to provide a preliminary reference for its performance.
>
> Results with MTR:
> | Methods        | Map Type | minADE $\downarrow$ | minFDE $\downarrow$ | MR $\downarrow$ |
> | -------------- | -------- | ------------------- | ------------------- | --------------- |
> | Offline Map    | HD map   | 0.3464              | 0.7396              | 0.0803          |
> | Offline Map    | SD map   | 0.3732              | 0.7732              | 0.0841          |
> | Online Mapping | HD map   | 0.3128              | 0.6892              | 0.0730          |
> | ESDMotion      | SD map   | 0.2883              | 0.6359              | 0.0672          |
>
> >**Q2. The proposed ESDMotion aims for end-to-end motion prediction however it still relies on SD maps as extra inputs, which is different from previous end-to-end works.**
>
> Indeed, the two approaches you mentioned are pioneering and valuable works in the field of end-to-end autonomous driving. The two approaches both propose to end-to-end conduct detection and motion. PIP additionally conducts online mapping.
>
> However, we would like to clarify that:
> - The cooperative relations of detection and motion prediction is not the focus of this work.  In this work, **our primary contributions are replacing HD Map with low-cost SD Map for motion prediction.**
> - The protocols of the two pioneering works introduce the influence of detection into the motion prediction results.  To study the influence of map in a decoupled way, we adopt the protocols of MapUncertainty (CVPR24 Best Paper Final List)/BEVPred (ECCV24) so that the conclusion would not be influenced by the detection module. End-to-end feature usage are our solutions for the issues caused by using SD maps.
> -  Compared to the ViP3D requring HD Map as inputs and PIP requiring HD Map for supervision, ESDMotion only requires the easily accessible SD Map (for example, Google Map), which could significantly reduce the cost.
>
> | Models                                    | Map Information            | Agents'  Information |
> | ----------------------------------------- | -------------------- | -------------------- |
> | [ViP3D](https://arxiv.org/pdf/2208.01582) | GT HD maps       | Detection results    |
> | [PIP](https://arxiv.org/pdf/2212.02181)   | Predicted HD maps | Detection results    |
> | ESDMotion                                 | GT/predicted SD maps      | GT                   |
>
> We add the above discussions to Appendix D of [our updated paper](https://openreview.net/pdf?id=sEJYPiVEt4) and thanks for mentioning the two important related works to better position this work.
>
>
> >**Q3. The use of SDMap feature fusion as a way to enhance BEV features is already common and lacks novelty.**
>
> Indeed, in the field of online mapping, there are already several works utilizing SD map as priors for HD map generation in 2024.
>
> However, we argue that our work has unique contributions:
> - We are the first to study the influence of directly using SD Map for motion prediction. The study aims to even not conducting HD map online mapping, which further reduces the cost.
> - We find out issues (low resolution and poor alignment) caused by using SD maps for motion prediction and propose tailored solutions for it.
> - By applying tailored designs, using SD maps could achieve comparable performance with using HD maps under fair settings, as shown in the updated Table 1.
>
> >**Q4. The nuScenes dataset mostly contains simple road structure scenarios (PARA-Drive, BEVPlanner), in some complex scenarios, will the ESDMotion scheme proposed by the authors perform more excellently, for example, compared with PARA-Drive?**
>
> This is an interesting point. Unfortunately, **PARA-Drive is not open-sourced yet (https://xinshuoweng.github.io/paradrive/), and BEVPlanner is designed for the planning task even without a motion prediction module, which differs from our motion prediction task**. Therefore, we are unable to directly compare our method with these approaches.
>
> **We agree that exploring the application of SD maps in more complex road scenarios, as well as their potential use in planning tasks, is an intriguing and worthwhile direction for future research**.  We add the above discussions to Appendix E of [our updated paper](https://openreview.net/pdf?id=sEJYPiVEt4) as future work.

---

> > ### Author Response · Authors · 2024-11-25
> >
> > Dear Reviewer,
> >
> > We hope our response regarding **New Baseline** solves your concerns.  If there is any further question, we are glad to discuss.
> >
> > Since the discussion period comes to end, regarding the novelty and contributions of this work, if you have any suggestions or concerns regarding **the major motivation of SD map oriented motion prediction**, please do not hesitate to discuss.
> >
> > Thanks,
> >
> >  Authors of Submission 2920

---

> > > ### Comment · Reviewer_Arah · 2024-11-28
> > >
> > > Thank the authors for the detailed response. My concerns have been covered, and I am inclined to accept the paper and keep my score.

---

> > > > ### Author Response · Authors · 2024-11-28
> > > >
> > > > Thanks for your reply and positive rating. We are glad that your concerns are solved and thanks for your kind advice to improve the manuscript.

---

### Official Review · Reviewer_NGwg · 2024-11-05

**Soundness:** 3
**Presentation:** 3
**Contribution:** 3
**Rating:** 5
**Confidence:** 5

**Summary:**

This paper proposes an end-to-end trajectory prediction approach based on SD-map, termed ESDMotion, and validates its effectiveness across two common trajectory prediction paradigms: anchor-based and anchor-free schemes. The core contributions include an Enhanced Road Observation strategy to facilitate interaction between agent and SD-map features, as well as a Pseudo Lane Expansion approach to address the insufficient sampling of goal points in anchor-based methods by broadening the sampling range. Experiments on the nuScenes dataset shows the performance improvements in trajectory prediction using SD-map, narrowing the gap with the strategies that rely on HD maps.

**Strengths:**

1.This paper is the first to propose and validate the use of SD-Map for end-to-end trajectory prediction tasks. It achieves performance that is comparable to, or even surpasses, approaches utilizing online predicted HD-map.
2.The proposed strategy effectively reduces the trajectory prediction performance gap between using SD-map and HD-map.

**Weaknesses:**

1.The Pseudo Lane Expansion strategy proposed is a relatively coarse goal point sampling approach, as the generated pseudo lanes may not align with the actual road structure, potentially resulting in out-of-bound instances and other inaccuracies.
2.In the experimental section, validation is lacking for the performance of replacing SD-map with online HD-map predictions from models such as MapTRV2, combined with the proposed strategies in this study. Including this experiment would greatly enhance the validity of the paper's findings.

Limitations

Although HD-maps are not required, this study still relies on offline-generated SD-Map, which limits its applicability. Additionally, for anchor-based trajectory prediction approaches, the proposed strategy does not demonstrate a performance advantage over methods that utilize online-predicted maps, highlighting a limitation of this approach.

**Questions:**

What will the trajectory prediction performance be like if the online predicted SD-map is used as input instead of the offline SD-map?

---

> ### Author Response · Authors · 2024-11-21
>
> Thanks for your acknowledgement and kind advice. Regarding your concerns, we give responses below:
>
> > **Q1&Q2. The Pseudo Lane Expansion strategy proposed is a relatively coarse goal point sampling approach, and may result in misalignment inaccuracies; The anchor-based trajectory prediction approaches does not demonstrate a performance advantage over methods that utilize online-predicted maps.**
>
> Thanks for your in-depth thinking. We agree that there could be potential misalignment of generated pseudo lanes with actual road structures. As a result, compared to predicted HD maps, the sparesity and misalignment of lanes in SD map would lead to missing of nearby candidates for anchor based methods like DenseTNT and thus worse performance.
>
> Inspired by you and other reviewers' thoughts, we improve DenseTNT under SD map scenario in two ways:
> 1. There are small areas like parking lots that are not covered by maps. For these areas, we define a set of pseudo lanes at specific angles capturing the potential driving directions to generate goal candidates, as shown in this [figure](https://postimg.cc/hhgXPL3b). This updated version is named DenseTNT+ and ESDMotion+. As shown in Table below, such modifications explicitly improve the performance of DenseTNT.
>
> | Models    | Maps      | minADE $\downarrow$ | minFDE $\downarrow$ | MR $\downarrow$ |
> | ----------| ---       | ------------------- | ------------------- | --------------- |
> | DenseTNT  | SD map | 1.2117              | 1.9849              | 0.2776          |
> | DenseTNT+ | SD map | 1.1746              | 1.9421              | 0.2630          |
> | ESDMotion | SD map | 0.7941              | 1.3863              | 0.1627          |
> | ESDMotion+| SD map | 0.7783              | 1.3784              | 0.1616          |
>
> 2. In the origin version of Pseudo Lane Expansion, we expand a fixed number of parallel lines at uniform distances, which introduces too many candidates and thus hurts performance. To reduce those unnecessary candidates, **we propose to dynamically adjust the parameters of Pseudo Lane Expansion based on the position and density of SD map.**
> - As shown in [figure](https://postimg.cc/qNZJsnLY) (upper part)/Figure 5 in our paper, SD lanes may align well with the road center but may also misalign and extend outside the drivable area.  In misaligned cases, the target vehicle is often farther from the SD lane. For misaligned cases, **we propose to generate more pseudo lanes on the side closer to the vehicle and reduce the number on the opposite side**, thereby minimizing the degree of misalignment.
> -  As shown in [figure](https://postimg.cc/qNZJsnLY) (lower part)/Figure 5 in our paper, the density of SD map is different in different areas. Thus, for those scenarios with sparse SD maps, we increase the number of pseudo lanes to ensure better coverage of the road surface. Conversely, in  areas with dense SD maps (e.g., intersections), we decrease the number and spacing of pseudo lanes to avoid overlap and interference. This further updated version is named ESDMotion++.
>
> | Models           | Maps   | minADE $\downarrow$ | minFDE $\downarrow$ | MR $\downarrow$ |
> | ---------------- | ------ | ------------------- | ------------------- | --------------- |
> | DenseTNT+BEVPred | HD map | 0.7630              | 1.3609              | 0.1576          |
> | ESDMotion        | SD map | 0.7941              | 1.3863              | 0.1627          |
> | ESDMotion+       | SD map | 0.7783              | 1.3784              | 0.1616          |
> | ESDMotion++      | SD map | 0.7597              | 1.3105              | 0.1523          |
>
> We add the above discussions and experiments to Section 2.3, Table 1, and Table 3 of [our updated paper](https://openreview.net/pdf?id=sEJYPiVEt4) and thanks for raising your concerns regarding anchor-based methods.

---

> > ### Author Response · Authors · 2024-11-21
> >
> > > **Q3. Add experiments about replacing SD-map with online HD-map predictions from models.**
> >
> > Thank you for your valuable suggestion. We conduct your suggested experiments by using HD maps predicted by models such as MapTR and MapTRv2-CL as inputs to ESDMotion. The tables above present the results on anchor-free model and anchor-based model.
> >
> > Results with HiVT:
> > | Maps             | minADE $\downarrow$ | minFDE $\downarrow$ | MR $\downarrow$ |
> > | ---------------- | ------------------- | ------------------- | --------------- |
> > | SD map(GT)       | 0.3134              | 0.6662              | 0.0737          |
> > | HD map (MapTR)   | 0.3147              | 0.6671              | 0.0740          |
> > | HD map (MapTRv2) | 0.3114              | 0.6692              | 0.0727          |
> >
> > Results with DenseTNT:
> > | Maps             | minADE $\downarrow$ | minFDE $\downarrow$ | MR $\downarrow$ |
> > | ---------------- | ------------------- | ------------------- | --------------- |
> > | SD map(GT)       | 0.7597              | 1.3105              | 0.1523          |
> > | HD map (MapTR)   | 0.7616              | 1.3139              | 0.1533          |
> > | HD map (MapTRv2) | 0.7529              | 1.3088              | 0.1517          |
> >
> >
> > **The results indicate that the performance of ESDMotion with predicted HD maps is similiar to that with SD maps**. This finding demonstrates that ESDMotion effectively reduces the performance gap between using SD maps and HD maps, validating the efficacy of the proposed SD map oriented designs.
> >
> > We add the above discussions and experiments to Table 1 of [our updated paper](https://openreview.net/pdf?id=sEJYPiVEt4) and thanks for raising your advice to better illustrate the advantages of ESDMotion.
> >
> >
> >
> > > **Q4. What will the trajectory prediction performance be like if the online predicted SD-map is used as input instead of the offline SD-map?**
> >
> > Good point and thanks for your advice!
> >
> > - We conduct experiments using the predicted SD map generated by MapTRv2 as input for four methods. **The results show that when using the predicted SD map, ESDMotion achieves performance similar to that with ground truth SD maps or predicted HD maps**. In contrast, for the base and unc baselines, using the predicted SD map leads to significant performance degradation. For the BEVPred baseline, performance with the predicted SD map is better than using ground truth SD maps alone but remains significantly lower than when using HD maps.
> >
> > - **The results demonstrate the effectiveness of the proposed designs for SD map oriented motion prediction**. It accounts for the relative inaccuracies inherent to SD maps and demonstrates strong robustness. This robustness effectively mitigates the impact of errors introduced by predicted SD maps, ensuring reliable performance under such conditions.
> >
> > Results with HiVT:
> > | Models           | MAPs              | minADE $\downarrow$ | minFDE $\downarrow$ | MR $\downarrow$ |
> > | ---------------- | ----------------- | ------------------- | ------------------- | --------------- |
> > | Base/Unc/BEVPred | GT SD map         | 0.3998              | 0.8207              | 0.0918          |
> > | Base             | Predicted SD map  | 0.4429              | 0.9165              | 0.0986          |
> > | Unc              | Predicted  SD map | 0.4285              | 0.9007              | 0.0956          |
> > | BEVPred          | Predicted  SD map | 0.3904              | 0.7690              | 0.0741          |
> > | ESDMotion++      | GT SD map         | 0.3114              | 0.6692              | 0.0727          |
> > | ESDMotion++      | Predicted  SD map | 0.3147              | 0.6671              | 0.0740          |
> >
> > Results with DenseTNT:
> > | Models           | MAPs              | minADE $\downarrow$ | minFDE $\downarrow$ | MR $\downarrow$ |
> > | ---------------- | ----------------- | ------------------- | ------------------- | --------------- |
> > | Base/Unc/BEVPred | GT SD map         | 1.2117              | 1.9849              | 0.2776          |
> > | Base             | Predicted SD map  | 1.3692              | 2.2417              | 0.3937          |
> > | Unc              | Predicted  SD map | 1.3020              | 2.1364              | 0.3738          |
> > | BEVPred          | Predicted  SD map | 1.1940              | 2.0029              | 0.3285          |
> > | ESDMotion++      | GT SD map         | 0.7597              | 1.3105              | 0.1523          |
> > | ESDMotion++      | Predicted  SD map | 0.7712              | 1.3260              | 0.1561          |
> >
> > We add the above discussions and experiments to Section 3.2 and Table 1 of [our updated paper](https://openreview.net/pdf?id=sEJYPiVEt4) and thanks for raising your advice to better illustrate the advantages of ESDMotion.

---

> > > ### Author Response · Authors · 2024-11-21
> > >
> > > > **Q5. Although HD-maps are not required, this study still relies on offline-generated SD-Map, which limits its applicability.**
> > >
> > > It is true that our method requires offline-generated SD maps as additional input. However, this type of information is relatively easy to obtain. For example, widely used navigation tools like **Google Map** provides SD maps that cover thousands of cities across most countries and regions. Compared with motion prediction without maps, leveraging such readily accessible data could save lots of cost of building and maintaining HD maps for autonomous driving companies.

---

> > > > ### Author Response · Authors · 2024-11-25
> > > >
> > > > Dear Reviewer,
> > > >
> > > > We hope our response regarding **Pseudo Lane Expansion** and **More Baselines** solves your concerns.  If there is any further question, we are glad to discuss.
> > > >
> > > > Since the discussion period comes to end,  we humbly request you to consider improving the score if you agree that the quality of the updated manuscript improves and satisfies the stanard of positive scores.
> > > >
> > > > Thanks,
> > > >
> > > >  Authors of Submission 2920

---

> > > > > ### Comment · Reviewer_TmWe · 2024-11-26
> > > > > **Thanks for the resonpose of authors**
> > > > >
> > > > > I have thoroughly read the author's response. I am glad that the author was able to address the issues I raised regarding DenseTNT. However, I still believe the contributions of this paper are currently insufficient to justify the acceptance of this paper at ICLR, a conference on learning algorithms.
> > > > >
> > > > > The main contribution of the paper is ESDMOTION, an end-to-end trajectory prediction model that utilizes SD maps and RGB images. However, considering the following points:
> > > > >
> > > > > 1.BEVPred can directly remove the supervision from HD map construction and perform the same end-to-end trajectory prediction based on BEV features as ESDMOTION.
> > > > >
> > > > >
> > > > > 2.Building on point 1, the BEV-SD interaction work in the HD map construction domain can incorporate SD priors into trajectory prediction. These works also identify and aim to solve the issues of low resolution in SD maps and the misalignment between SD and HD maps.
> > > > >
> > > > > Therefore, the contribution of this paper is essentially limited to using specific rule-based expansions of SD instances to improve the effectiveness of SD maps in trajectory prediction. I admit that these rule-based expansions is novel in my understanding.

---

> ### Author Response · Authors · 2024-11-26
>
> Thanks for your response and careful thinking!  We are glad that we could discuss more regarding our contributions:
>
> > **BEVPred can directly remove the supervision from HD map construction and perform the same end-to-end trajectory prediction based on BEV features as ESDMOTION.**
>
> - First, we agree that end-to-end is not our contribution, which has been done in BEVPred.
>
> - However, **our end-to-end design is tailored for SD Map**. For example, as shown in Fig. 4, Equ. 5, and L263-L283, we design a deformable attention mechanism - *Enhanced Road Observation*, where it first deformably attends  to all potential parallel areas and then fuses these information with predicted confidence h_i for those different areas. **This deformable attention is specifically designed for the sparse SD map lanes so that it could adaptively find out meaningful areas**. The proposed methodology is beyond the scope of BEVPred.
>
> > The BEV-SD interaction work in the HD map construction domain can incorporate SD priors into trajectory prediction. These works also identify and aim to solve the issues of low resolution in SD maps and the misalignment between SD and HD maps.
>
> - We agree that there are multiple works combines SD Map to generate HD Map where **their focus is on utilizing the SD Map as strong priors for HD map generation.**
>
>  - Regrading *These works also identify and aim to solve the issues of low resolution in SD maps and the misalignment between SD and HD maps*, we argue that **these issues are actually  motion prediction specific**, caused by the removal of reliance on HD map in motion prediction. Thus, our methodologies are different from online mapping methods as we focus on the downstream task - motion prediction. As mentioned in your summary, our high level goal is to  expand SD instances to avoid the missing of map elements when predicting agents' future trajectories. We aim to let the motion prediction model takes these noisy expansion as inputs and adaptively selects useful information aided by end-to-end learning.
>
> > **The contribution of this paper is essentially limited to using specific rule-based expansions of SD instances to improve the effectiveness of SD maps in trajectory prediction**.
>
> The reviewer indeed has very in-depth thinking of our methodology and we appreciate this point. We agree that the high level inspirations of all of our technical designs are to expand the SD map lane as they are sparse and sometimes  inaccurate.
>
> However, **we argue that our designs should not be understood as rule-based post-processing methodologies**.  Enhanced Road Observation utilizes attention mechanism to find out meaningful ares. Pseudo Lane Expansion and Adaptive Pseudo Lane Expansion (Please check supplymental Section C) change the training process for anchor-based methods like DenseTNT under usage of SD Map.  All of designs are innovations of learning algorithms.  We aim to design proper ways to fuse the noisy expanded SD instances and we design end-to-end learning modules to do it.
>
> > **I still believe the contributions of this paper are currently insufficient to justify the acceptance of this paper at ICLR, a conference on learning algorithms.**
>
> Beyond the technical contributions mentioned above, we would like to also discuss about the types of contributions one work could have.  **We argue that taking the initial steps to explore new possibilites are as equally important as methodology innovations.** For example,  the pioneering work Unc[I] explores the setting of adopting online mapping models for motion prediction.  It gives in-depth analysis regarding the issues of online mapping models and proposes an simple yet effective solution - uncertainty.   From the perspective of learning algorithms, the concept of uncertainty is not new. However, it shed insights regarding the topic and thus even entered CVPR24 best paper final list.
>
> **ESDMotion is the first to explore the possibilities of replacing HD Map with SD Map for motion prediction.** Following your and other reviewers' kind advice, we conduct detailed experiments to compare the all potential settings of SD Map VS HD Map.  We believe these findings are also insightful contributions, worth sharing in the community.
>
> With all due respect, we deeply appreciate your in-depth thinking and your willingness of taking your valuable time to discuss about our contributions.
>
> Thanks,
>
> Authors of Submission 2920
>
>
> [I]  Producing and Leveraging Online Map Uncertainty in Trajectory Prediction. CVPR 2024

---

> > ### Author Response · Authors · 2024-11-29
> >
> > Dear Reviewer,
> >
> > As the last day that reviewers may post a message is December 2nd, we are wondering whether your remaining concerns are solved and we humbly request you to consider a positive score, especially we noticed your acknowledgement regarding our designs. As the first work to discuss about applying SD map into motion prediction, we believe that our findings and proposed designs have enough contributions, worth sharing in the community.
> >
> > Thanks
> >
> > Authors of Submission 2920

---

> > > ### Author Response · Authors · 2024-12-02
> > >
> > > Dear Reviewer,
> > >
> > > **Since we have responded your existing concerns with solid experiments and validations and there is no new concern raised**, we humbly request you to consider adjusting scores accordingly. If there are any further concerns, we are glad to respond.
> > >
> > > Thanks
> > >
> > > Authors of Submission 2920

---

### Author Response · Authors · 2024-11-21
**General Response**

Dear AC and reviewers,

We express our gratitude to all reviewers for their valuable time and insightful comments.
Following your kind advice, we add more experiments, results, and explanations. Please check [our updated paper](https://openreview.net/pdf?id=sEJYPiVEt4) where the changes are marked in blue.

There are two common concerns raised and we clarify below.

> **Q1. End-to-end learning for motion prediction is not new. What is the contribution of this work?**

- The major focus of this work is not about end-to-end learning. In this work, we aim to address a previously unexplored problem: **using SD maps as the sole map information for motion prediction to reduce the cost of relying on HD maps**.
- We find out two issues (low resolution and poor alignment) caused by using SD maps for motion prediction. To overcome these issues, we design an end-to-end framework and tailored modules to better utilize SD maps. **The end-to-end learning is the high-level methodology of our solutions while our contributions lie in the SD map specific designs**.
-  As a result, our method demonstrates that **even with the low-cost SD maps, it is possible to achieve motion prediction performance comparable to that using HD maps**.
- As pointed out by multiple reviewers, we agree that current titles and writing may cause misunderstanding regarding the contributions. Thus, following reviewers' kind advice, we change our title to **ESDMotion: SD Map Oriented Motion Prediction Enhanced by End-to-end Learning** to emphasize our focus of SD map for motion. Accordingly, we revise the structure and wording in [our updated paper](https://openreview.net/pdf?id=sEJYPiVEt4) to emphasize the SD map oriented motivations.

> **Q2. Why do anchor-based methods fail to achieve the same level of improvement as anchor free method?**

The reason is that: **Anchor-based methods like DenseTNT use lanes to generate goal candidates**. The sparesity and misalignment of lanes in SD map would lead to missing of nearby candidates and thus bad performance.

Inspired by reviewers' advice, we improve DenseTNT under SD map scenario in two ways:
1. There are small areas like parking lots that are not covered by maps. For these areas, we define a set of pseudo lanes at specific angles capturing the potential driving directions to generate goal candidates, as shown in this [figure](https://postimg.cc/hhgXPL3b). This updated version is named DenseTNT+ and ESDMotion+. As shown in Table below, such modifications explicitly improve the performance of DenseTNT.

| Models    | Maps      | minADE $\downarrow$ | minFDE $\downarrow$ | MR $\downarrow$ |
| ----------| ---       | ------------------- | ------------------- | --------------- |
| DenseTNT  | SD map | 1.2117              | 1.9849              | 0.2776          |
| DenseTNT+ | SD map | 1.1746              | 1.9421              | 0.2630          |
| ESDMotion | SD map | 0.7941              | 1.3863              | 0.1627          |
| ESDMotion+| SD map | 0.7783              | 1.3784              | 0.1616          |

2. In the origin version of Pseudo Lane Expansion, we expand a fixed number of parallel lines at uniform distances, which introduces too many candidates and thus hurts performance. To reduce those unnecessary candidates, **we propose to dynamically adjust the parameters of Pseudo Lane Expansion based on the position and density of SD map.** Specifically, for misaligned SD map as shown in [figure](https://postimg.cc/qNZJsnLY) (upper part), we generate more pseudo lanes on the side closer to the vehicle and reduce the number on the opposite side. To deal with different density of SD lanes as shown in [figure](https://postimg.cc/qNZJsnLY) (lower part)/Figure 5 in our paper, we increase the number of pseudo lanes to ensure better coverage for spare ones while decrease the number for dense ones. This further updated version is named ESDMotion++. As shown below, the performance is further enhanced.

| Models      | Maps   | minADE $\downarrow$ | minFDE $\downarrow$ | MR $\downarrow$ |
| ----------- | ------ | ------------------- | ------------------- | --------------- |
| ESDMotion   | SD map | 0.7941              | 1.3863              | 0.1627          |
| ESDMotion+  | SD map | 0.7783              | 1.3784              | 0.1616          |
| ESDMotion++ | SD map | 0.7597              | 1.3105              | 0.1523          |

We add these discussions, methods, and experiments into [our updated paper](https://openreview.net/pdf?id=sEJYPiVEt4). We sincerely thank the reviewers for their constructive comments.

We hope our responses could solve your concerns. If you have any further concerns or questions to discuss, we are more than willing to address them. Looking forward to your further suggestions to polish the paper!

Thanks,
Authors of Submission 2920

---

### Meta-Review · Area_Chair_Et6d · 2024-12-22

**Metareview:**

This paper presents a SD map conditioned motion prediction framework, which solely relies on SD maps and onboard signals. This paper reveals the gap between the models trained with SD maps and those with HD maps.  After reviewing the extensive discussions among the reviewers and authors, there is still a mix of borderline evaluations. Below, I summarize the key considerations.

Strengths:

1. This paper presents a novel setting that could benefit future autonomous driving pipelines.
2. The proposed methods are general, working with both anchor-based and anchor-free methods.
3. The results with SD maps are comparable to those with HD maps under several settings.
4. Thorough response and extensive additional experiments are provided in the rebuttal period.

Weaknesses:

1. The proposed method is not technically novel and similar ideas have been studied in prior arts.
2. Reviewers suggested experiments to combine SD maps and online predicted maps. Although authors provided additional experiments, however, reviewers suggested potential unfairness if authors did not include the aforementioned experiments.
3. Paper presentation quality has to been improved.

After reading these divided reviews and discussing with reviewers, AC decided to reject this paper because:

1. The paper's technical novelty is limited given the prior work suggested by reviewers. Also the empirical results do not justify the usage of SD maps due to the remaining gap. Both aspects do not meet the standard of ICLR.
2. Writing needs to be greatly improved to make the cut of ICLR.

**Additional Comments On Reviewer Discussion:**

This paper went through a long rebuttal. Initially, reviewers questioned the overall contributions and suggested multiple experiments to justify design choices, such as:

1. Why did anchor-based methods fail to achieve similar performance to anchor-free counterparts?
2. Lack of recent baselines. Selected baselines (e.g. DenseTNT) are too old.
3. What are the major contributions of this work?

After the rebuttal, most concerns regarding the design choices are addressed. However, reviewers (TmWe, 9KQE) still believe this paper's contributions are blurry and it fails to address all reviewers' concerns. Also, an experiment that combines SD maps and online map prediction is not provided. Therefore, there are several remaining questions to be addressed in a future submission.

---

### Decision · Program_Chairs · 2025-01-22

Reject